# CNS-Bench: Benchmarking Model Robustness Under Continuous Nuisance Shifts

## Abstract

One important challenge in evaluating the robustness of vision models is to control individual nuisance factors independently. While some simple synthetic corruptions are commonly applied to existing models, they do not fully capture all realistic distribution shifts of real-world images. Moreover, existing generative robustness benchmarks only perform manipulations on individual nuisance shifts in one step. We demonstrate the importance of gradual and continuous nuisance shifts, as they allow evaluating the sensitivity and failure points of vision models. In particular, we introduce **CNS-Bench**, a **C**ontinuous **N**uisance **S**hift **Bench**mark for image classifier robustness. CNS-Bench allows generating a wide range of individual nuisance shifts in continuous severities by applying LoRA adapters to diffusion models. After accounting for unrealistic generated images through an improved filtering mechanism for such samples, we perform a comprehensive large-scale study to evaluate the robustness of classifiers under various nuisance shifts. Through carefully-designed comparisons and analyses, we find that model rankings can change for varying shifts and shift scales, which is not captured when averaging the performance over all severities. Additionally, evaluating the model performance on a continuous scale allows the identification of model failure points, providing a more nuanced understanding of model robustness. Overall, our work demonstrated the advantage of using generative models for benchmarking robustness across diverse and continuous real-world nuisance shifts in a controlled and scalable manner.

## 1 Introduction

Machine learning models are typically validated and tested on fixed datasets under the assumption of independent and identically distributed samples. This, however, does not fully cover the true capabilities and potential vulnerabilities of models when deployed in dynamic real-world environments. The robustness in out-of-distribution (OOD) scenarios is important and decision-makers might need to know how models perform under various distribution shifts and severity levels in safety-critical scenarios. Therefore, it is crucial to continue building richer and more systematic benchmarks.

In the past few years, various benchmarks have been proposed to evaluate the robustness of computer vision models. One line of benchmarks manually collects data with nuisance shifts (Zhao et al., 2022; Hendrycks et al., 2021a; Wang et al., 2019; Geirhos et al., 2022; Barbu et al., 2019; Idrissi et al., 2022; Hendrycks et al., 2021b; Recht et al., 2019). Yet, such approaches are not scalable and often include only a small variety of nuisance shifts.

On the other hand, synthetic datasets offer opportunities to evaluate deep neural networks since various instances of an object class with specified context and nuisance shifts can be generated. While rendering pipelines allow precise control of several variables and are applied for benchmarking (Bordes et al., 2024; Shu et al., 2020; Kar et al., 2022; Li et al., 2023c), some nuisance shifts such as weather variations (*e.g.*, snow) are very hard to perform using traditional pipelines. While Hendrycks & Dietterich (2018) report accuracy drops for various types and levels of synthetic corruptions, they lack relevant real-world nuisance shifts.

Recent developments in diffusion models have enabled the application of generative models for training (He et al., 2022b; Fan et al., 2024) and benchmarking vision models (Mofayezi & Medghalchi, 2023; Metzen et al., 2023; Vendrow et al., 2023; Zhang et al., 2024). However, all

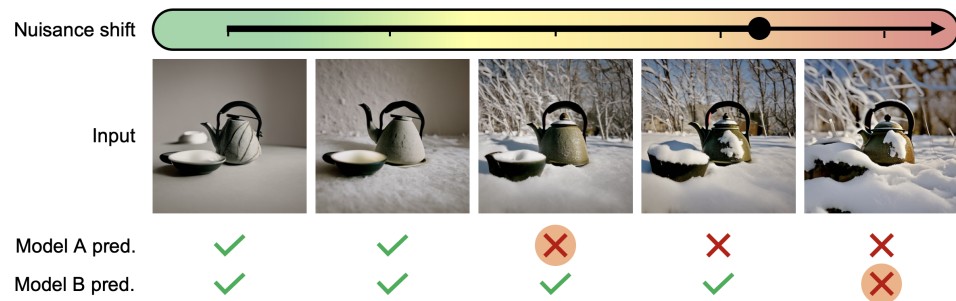

Figure 1: **Benchmarking under continuous nuisance shifts.** We evaluate the robustness of different models under gradually increasing nuisance shifts. This allows identifying the *failure point* (highlighted in red) of a model.

previous approaches define *categorical* or *binary* nuisance shifts by considering the existence or absence of a shift, which contradicts their continuous realization in real-world scenarios. For example, as shown in Fig. 1, the snow level in an environment can range from light snowfall to objects fully covered with snow. While one model might fail at all snow levels, a different model might only fail when the object is heavily occluded. In most real-world applications, it is important to know the expected performance at specific nuisance shift levels, rather than just a global accuracy drop. For instance, an autonomous driving company may need to determine the fog density at which system performance falls below a critical threshold. Evaluating such failure points to probe the sensitivity of models requires realizing continuous shifts.

To overcome this shortcoming, we establish a **C**ontinuous **N**uisance **S**hift **Bench**mark for model robustness, dubbed as **CNS-Bench**. Specifically, we apply LoRA (Hu et al., 2021) adapters to diffusion models to perform a continuous variation of specified nuisance shifts, and use them to benchmark a variety of classifiers along the following axes: (i) architecture, (ii) number of parameters, (iii) pre-training paradigm and data. In contrast to previous works conducting analysis on *binary* or *categorical* shifts, our study advocates multiple scales of shifts. We caveat that model rankings can change when considering several scales. It is also essential to consider failure points, *i.e.*, the shift severity at which a model fails. Thus, measuring robustness as a spectrum instead of aggregating it into a single average metric allows a more comprehensive understanding of OOD robustness (Drenkow et al., 2021; Hendrycks et al., 2021a). With our benchmark, we evaluate more than 40 classifiers and demonstrate that a rigorously-designed generative benchmark allows systematically studying the robustness behaviors of vision models in a controlled and scalable manner.

One essential requirement when using synthetic images for benchmarking is to ensure that the considered images correspond to the class distribution. Manually checking the quality of images to find those not aligned with the desired condition is still a common practice (Zhang et al., 2024). However, it has difficulty in scaling up the analysis (Hastie et al., 2009; Angelopoulos et al., 2023). Some approaches have been proposed for automatic filtering, but no standard datasets are available to evaluate filtering strategies. With this in mind, we also provide a dataset with manually annotated out-of-class (OOC) images. We show that our proposed filtering mechanism outperforms previous strategies in removing such problematic samples.

In summary, our work makes the following contributions: **1)** We propose CNS-Bench to benchmark vision models under continuous nuisance shifts. We publish a dataset with 14 diverse and realistic nuisance shifts that represent various style and weather variations at five severity levels. In addition, we also provide trained LoRA sliders for all shifts that can be used to compute shift levels in a fully continuous manner. **2)** We collect an annotated dataset to benchmark OOC filtering strategies and propose a novel filtering mechanism that achieves higher filter accuracies than previous methods. **3)** We evaluate the robustness of more than 40 classifiers along different axes and reveal multiple valuable findings, underlining the importance of considering continuous shift severities of real-world nuisance shifts.

## 2 RELATED WORK

**Robustness.** When referring to robustness, we consider the relative accuracy drop of a classifier *w.r.t.* interventions that alter images from a base distribution, building upon the formalism introduced in Drenkow et al. (2021). While the averaged accuracy drops provide an aggregated measure of the robustness, we consider the robustness *w.r.t.* specific nuisance shifts that can be modeled as causal interventions on the environment, the appearance, the object, or the renderer. We define such continuous interventions on metric scale.

**Benchmarking robustness.** Early approaches for benchmarking the performance and generalizability of models use fixed datasets, assuming independent and identically distributed samples (Deng, 2012; Deng et al., 2009; Lin et al., 2014). However, this lacks scalability and fails to capture the performance in real-world applications facing OOD scenarios. To tackle this challenge, a line of research involves manually collecting data with nuisance shifts (Zhao et al., 2022; Hendrycks et al., 2021a; Wang et al., 2019; Geirhos et al., 2022; Barbu et al., 2019; Idrissi et al., 2022; Hendrycks et al., 2021b; Recht et al., 2019). However, these methods are often time-consuming and labor-intensive since they require data crawling and human annotations. Moreover, they usually capture only a subset of nuisance shifts that models may encounter in the real world and it is challenging to ensure the disentanglement of these annotated nuisances.

Another line of research uses synthetic data for benchmarking, which offers the ability to generate a large and diverse range of nuisance shifts with precise control (Hendrycks & Dietterich, 2018; Bordes et al., 2024; Shu et al., 2020; Kar et al., 2022). However, these works are limited to nuisances that can be easily modelled (*e.g.*, lighting, fog, occlusions) or restricted to what can be expressed in rendering pipelines. Recent developments in diffusion models shed light on creating realistic and diverse synthetic benchmark datasets (Mofayezi & Medghalchi, 2023; Metzen et al., 2023; Vendrow et al., 2023; Zhang et al., 2024) with realistic data and more possibilities to control nuisances (*e.g.*, text-guided corruptions, counterfactual). In our work, we propose a framework to benchmark vision models *w.r.t.* nuisance shifts under multiple severity levels. To address the need to remove OOC images from generative models, which are essential for benchmarking applications, we additionally propose a novel strategy to remove such samples from the dataset.

## 3 CONTINUOUS NUISANCE SHIFT BENCHMARK

In this section, we present how CNS-Bench is created. We first discuss the strategy to replicate the in-domain distribution in Section 3.1. We then present our methodology to perform continuous shifts to evaluate the model's sensitivity to various nuisance factors in Section 3.2. Finally, we detail our filtering dataset and the selected filtering strategy in Section 3.3.

### 3.1 REPLICATING THE IMAGENET DISTRIBUTION

We aim to evaluate a model's robustness to specific nuisance shifts that alter the base ImageNet (Deng et al., 2009) distribution $p(X_{\text{IN}}|c)$, conditioned on an ImageNet class $c$. However, as pointed out by Vendrow et al. (2023), the distribution of Stable Diffusion (SD) (Rombach et al., 2022) generated images $p(X_{\text{SD}}|c)$ differs from the ImageNet distribution, significantly lowering classification accuracies. To generate images that are more similar to the ImageNet images, we apply textual inversion (Gal et al., 2023) to learn new "words" in the embeddings space of a text encoder that capture the ImageNet-specific class concepts. Specifically, these text embeddings are optimized by minimizing the noise prediction error of diffusion models $||\epsilon - \epsilon_\psi(\cdot, f_\psi(c)||^2$ with the text encoder $f_\psi(\cdot)$ and parameters $\psi$ for all diffusion time steps. We call this distribution IN*: $p(X|c) = p(X_{\text{IN*}}|c)$.

### 3.2 CONTINUOUS NUISANCE SHIFTS FOR BENCHMARKING

To evaluate the robustness of vision models *w.r.t.* continuous nuisance shifts, the following characteristics are desirable: (i) The severity of the considered shift can be controlled, allowing the estimation of the shift scale where a considered model fails. (ii) Realizing a nuisance shift should not come along with variations that might alter the class identity. (iii) The variations should be subtle, allowing a fine-grained analysis also for specific images.

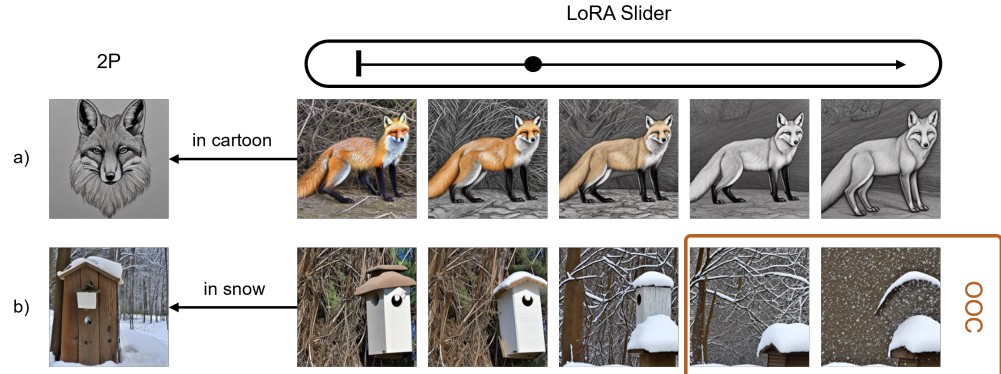

Figure 2: **Qualitative examples for prompt-based and LoRA-based shifts with out-of-class samples.** On the left, we present two images in a different a) style and b) weather condition generated from a text prompt a) "fox in cartoon style" and b) "birdhouse in heavy snow", respectively. On the right, we show the gradual variation performed by our LoRA sliders. a) Unlike the prompt-based shift, our LoRA sliders successfully generated images showing a gradual shift. b) Our LoRA sliders sometimes result in out-of-class (OOC) samples for higher scales, as depicted with the orange box.

**Realizing continuous nuisance shifts.** A natural way to perform synthetic nuisance shifts are methods based on text prompts (Metzen et al., 2023; Liu et al., 2023; Vendrow et al., 2023). They follow the two prompt (2P) templates: "A picture of a <class>" and "A picture of a <class> in <shift>". However, this approach does not allow the gradual increase of a nuisance for a given image. In addition, the generated shifts largely vary for different seeds and classes when applying the prompt addition "in <shift>"—for some seeds, the generated shift is more prominent, while for others, it is barely visible. Additionally, the semantic structure of the generated image can be significantly changed.

We leverage LoRA (Hu et al., 2021) adapters that represent low-rank matrices added to the original weight matrices to perform continuous shifts. Such adapters are trained to characterize the effect of a considered nuisance shift. Gandikota et al. (2023) propose a strategy to learn concept sliders using LoRA adapters that allow a continuous modulation of the considered concept, which is achieved by learning low-rank matrices that increase the expression of a specific attribute when applied to a class concept $c$. The low-rank parameters $\theta_{\mathrm{LoRA}}$ modify the original model parameters $\theta$ to $\theta^* = \theta + s \cdot \theta_{\mathrm{LoRA}}$ with scale $s$ are trained to capture a concept of interest $c_+$: $P_{\theta^*}(X|c) \leftarrow P_\theta(X|c) \cdot P_\theta(X|c_+)^\eta$, where $\eta$ refers to weighting factor that is fixed during training. Following Gandikota et al. (2023), we optimize with the MSE objective (Sohl-Dickstein et al., 2015) using the Tweedie's formula (Efron, 2011) and the reparametrization trick (Ho et al., 2020) by formulating the scores as a denoising prediction $\epsilon(X, c, t)$ with the diffusion timestep $t$: $\mathrm{MSE}(\epsilon_{\theta^*}(X, c, t); \epsilon_\theta(X, c, t) + \epsilon_\theta(X, c_+, t))$. We model the class concept $c$ and the nuisance concept $c_+$ by two text embeddings "<class>" and "<class> in <shift>". Different to (Gandikota et al., 2023), we specifically use class concepts $c$ that are acquired from the IN* distribution. After training, the learned LoRA adapters capture the direction between the two language concepts, i.e., they characterize attributes of the concept of interest $c_+$. Weighting their effect using the scale $s$ modulates the effect of the applied shift. Gandikota et al. (2023) stated that the LoRA adapters generalize to other concepts and images. We found that learning class-specific LoRA sliders produces higher-quality shifts. This choice also allows capturing the class-specific characteristics and confounders of the considered shifts that occur in the real world. Hence, we train separate LoRA adapters for each ImageNet class and shift. As qualitatively shown in Fig. 2, applying these learned directions enables gradual nuisance shifts. We show examples of more shifts in Fig. 33 and Fig. 34.

Following Mokady et al. (2023); Gandikota et al. (2023), we evaluate the shift severity based on the CLIP similarity of the generated image to the text prompt describing the shift, *i.e.*, "A picture in <shift>". Similarly, we also compute the CLIP (Radford et al., 2021) similarity to the class prompt "A picture of a <class>". To measure the performed shift, we compute the CLIP shift difference by $\Delta\mathrm{CLIP}_{\mathrm{shift}}(I_k, I_0) = \cos(\mathrm{CLIP}_{\mathrm{img}}(I_k), \mathrm{CLIP}_{\mathrm{text}}(\text{"in \{shift\}"})) - \cos(\mathrm{CLIP}_{\mathrm{img}}(I_0), \mathrm{CLIP}_{\mathrm{text}}(\text{"in \{shift\}"}))$ for the generated image with scale 0 and scale $k$, and similarly for the class similarity.

In contrast to simply applying a second text prompt (2P) to perform a *binary* shift, our LoRA adapters allow performing a variety of shift scales, as measured by the CLIP shift difference (Section 3.2). This allows gradual shifts (Fig. 2).

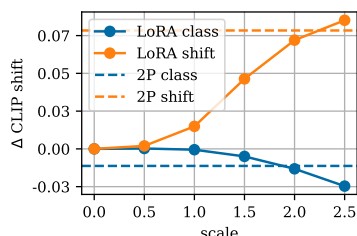

Activating the LoRA adapter at different time steps throughout the diffusion process will modulate the effect of the adapter on the generation process (Meng et al., 2021). If the LoRA adapter is active for all noise steps, it will significantly influence the semantic structure and the appearance of the generated image, while deactivating the adapter for earlier time steps will keep the semantic structure. Since we aim to perform more fine-grained edits that do not heavily change the semantic structure, we deactivate the LoRA adapter for early steps. This allows applying edits where the semantic structure remains similar but the appearance changes (*e.g.* Fig. 2a).

Figure 3: **Average $\Delta$ CLIP evaluation for the snow shift.** Our sliders perform a gradual shift, while a naive application (2P) only allows *binary* shifts.

Since our sliders do not explicitly exclude confounding variables, the applied shifts may also affect confounders inherently present due to biases in the training data. For example, as shown in Fig. 34c, using the *in dust* slider unintentionally removes half of the people, and in Fig. 33c, the background no longer represents a forest. Consequently, failures in our subsequent analysis cannot always be solely attributed to the nuisance concept itself.

**Failure point concept.** We define a failure point $s = \min\{S \in \mathbb{R} | f(X(S)) \neq c\}$ as the smallest shift scale where a classifier $f(X(s))$ fails to correctly classify an image $X(s)$ with a class $c$ and a scale $s$ of a considered shift. The failure point distribution captures the ratio of failed samples for the considered scales. We estimate this distribution in our work with a histogram, where the number of elements in one bin $I_k$ is computed by $H(I_k) = \sum_{n=1}^{N} \mathbb{1}_{I_k}(s_n)$ with the indicator function $\mathbb{1}(\cdot)$ and the scale of the $n$-th element of the set of images with $N$ images. We compute and report the ratio of failure points for each scale $s$, dividing $H(I_k)$ by the number of considered images $N$.

### 3.3 FILTERING DATASET AND STRATEGY

To evaluate filtering strategies for removing out-of-class (OOC) samples, we collect a manually labeled dataset. This section presents this dataset and the selected filtering strategy.

**Filtering of OOC samples.** Current diffusion models allow the generation of diverse and realistic images $x \sim p(X|\mathbf{z})$ that are conditioned on $\mathbf{z} = [c, s_i]$, which involves the considered ImageNet class $c \in \mathbb{N} \mid 1 \leq c \leq 1000$ and the variable $s_i \in \mathbb{R}$ corresponding to the severity of a considered nuisance shift $i$. However, due to their probabilistic formulation, the generated sample might deviate from the condition $\mathbf{z}$. For benchmarking applications, we are particularly concerned about generated samples deviating from the original class $c$, *i.e.*, the considered class cannot be characterized anymore (c.f., Fig. 2). We call such samples "OOC" samples (Metzen et al., 2023). Evaluating the sensitivity to specific nuisance shifts requires removing the OOC samples generated by the shift's application. Therefore, we collect a dataset of generated images to evaluate the sliding process and strategies to automatically remove OOC samples.

**Dataset for evaluating OOC filtering strategies.** To evaluate various OOC filtering strategies, we manually label a dataset consisting of 18k generated images with two shifts, five scales, and 100 random ImageNet classes. We select *snow* as one weather variation and *cartoon* as one style shift to represent two rather different nuisance shifts. Before manually labeling the dataset, we remove easy samples that have a high CLIP text alignment and are classified correctly by multiple classifiers. Then, all hard images are labeled by two human annotators, where each annotator can choose from the following labels: "class", "partial class properties", and "not class". More details on the labeling strategy and the dataset statistics are provided in Appendix A.6.

**OOC filtering strategy.** A filter serves its purpose if it removes all OOC samples, corresponding to a high true positive rate (TPR), while not removing too many in-class samples, corresponding to a low false positive rate (FPR). Instead of simply applying a CLIP threshold as in Vendrow et al. (2023), we consider a combinatorial selection approach, which requires two out of four filters to be active. For the first and the second filter, we consider text alignment to "A picture of a <class>"

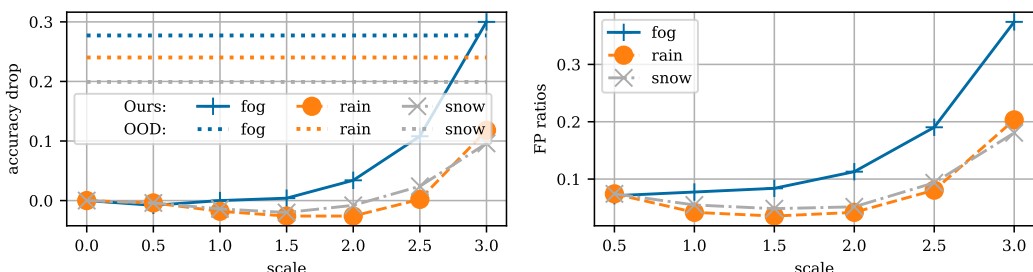

Figure 4: **Accuracy drops and failure point ratios of a ResNet-50 classifier on OOD-CV and our benchmark.** *Left*: Accuracy drops on OOD-CV and various scales of our benchmark (in the value range [0,1]). Horizontal lines show the average score for each weather nuisance of the OOD-CV test dataset, while our benchmark allows identfying the performance drop at various shift scales. *Right*: Distribution of failure points. Our continuous nuisance shifts allow identifying the scales that result in a failure and models fail earlier for fog, potentially due to heavier occlusions than snow and rain.

and "A picture of a <class> in <shift>", respectively, computed via CLIP. For the third and fourth filter, we measure the cosine similarity to the starting images using the CLIP image encoder and the class tokens of DINOv2 (Oquab et al., 2023), respectively. We select the filtering threshold for each filter such that 90% of the labeled OOC samples are removed. Note that none of these filters are trained on ImageNet data.

## 4 EXPERIMENTS

In this section, we discuss our benchmark results. First, we compare our bechmarking strategy with the OOD-CV benchmark. Then, we perform a large-scale analysis by evaluating more than 40 ImageNet classifiers on CNS-Bench.

### 4.1 COMPARING CONTINUOUS SHIFTS WITH OOD-CV DATASET

Zhao et al. (2022; 2024) introduce OOD-CV to measure out-of-distribution (OOD) robustness, a benchmark dataset that includes OOD examples of ten object categories for five different individual nuisance factors (*e.g.*, weather) on real data. OOD-CV is the only real-world dataset that provides accurate labels of various individual weather shifts. This allows comparing our generated images with real-world weather realizations of the considered shifts. We use our trained LoRA adapters to create a benchmark for the OOD-CV classes and scales up to $3.0$ to directly compare with the original manually labeled dataset. We refer to the supplementary for exemplary images of both benchmarks and CLIP alignments to the considered shifts.

First, we train a ResNet-50 classifier on the training set of the OOD-CV benchmark. Then, we evaluate the performance on our data and the OOD-CV benchmark. Fig. 4 presents the results for each nuisance independently. The accuracies remain more or less constant with an accuracy around $95\%$ up to a nuisance scale of $1.5$. From a nuisance scale of $2.0$, the accuracy starts dropping, with the nuisance of *fog* having the biggest impact. This could be explained by the fact that fog can lead to severe occlusion, while rain and snow can be considered as corruption factors. We hypothesize that the partially bigger drop for the OOD-CV benchmark is due to a major limitation of its dataset: The nuisances are not completely disentangled, and part of the accuracy drop originates from various other factors (*e.g.*, image quality, image size, and noise). In contrast, our benchmark allows for fine-grained control of nuisances with multiple shift levels, leading to a more complete and scalable analysis of the model's performance.

### 4.2 EVALUATED MODELS AND EXPERIMENTAL SETUP

We use our large-scale benchmark to evaluate the models along the following axes:
(i) *Architecture.* To compare architectures with a comparable number of parameters, we consider both CNN and ViT architectures with different training recipes: ResNet-152 (He et al., 2016), ViT-B/16 (Dosovitskiy et al., 2020), DeiT-B/16 (Touvron et al., 2021), DeiT-3-B/16 (Touvron et al.,

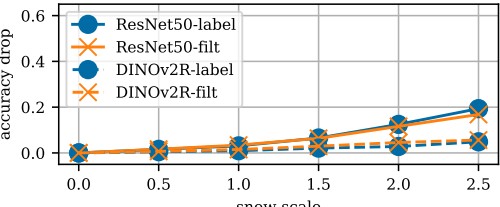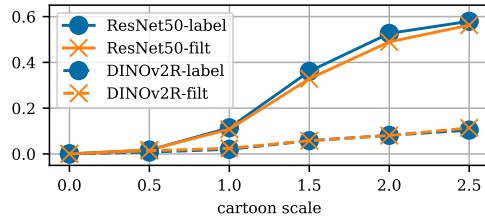

Figure 5: **Classification accuracy drops on the labeled and filtered datasets.** The accuracy drop curves of ResNet-50 and DINOv2 classifiers on the filtered and the labeled dataset are comparable, demonstrating the effectiveness of our automatic filtering strategy. The ratio of misclassified images is given in the value range [0,1]. We provide results for more classifiers in Fig. 8.

2022), and ConvNeXt-B (Liu et al., 2022). All models are trained in a supervised manner.

(ii) *Model size.* For ViT, we consider the small, medium, base, large, and huge variants of DeiT-3. For CNN, we consider the ResNet variants: 18, 34, 50, 101, and 152.

(iii) *Pre-training paradigm and data.* We evaluate a set of models with the same backbone but different pre-training strategies. The following models are pre-trained on IN1k with a self-supervised objective: MAE (He et al., 2022a), DINOv1 (Caron et al., 2021), and MoCov3 (Chen et al., 2021). To study the impact of more data during training, we compare their performance to a supervised model that is trained only on ImageNet-1k and a supervised model that is pre-trained on ImageNet-21k. All Transformer-based models use ViT-B/16 as the backbone. Furthermore, we evaluate an ImageNet-trained diffusion classifier (Li et al., 2023b) on a smaller subset due to its heavy computational cost.

**Metrics.** We typically report the average accuracy drops, i.e., the ratio of failed images, averaged over the images of one shift or all shifts in the value range $[0, 1]$. In Table 1, we report the mean relative corruption error (rCE) as introduced by Hendrycks & Dietterich (2018). It is defined by the average over all relative corruption errors $\text{CE}_{\text{shift}} = \left( \sum_s E_{\text{shift},s}^f - E_{\text{shift},0}^f \right) / \left( \sum_s E_{\text{shift},s}^{\text{alex}} - E_{\text{shift},0}^{\text{alex}} \right)$ with the average error $E$ for scale $s$, model $f$, and a specific shift.

**Slider details.** As pointed out in Section 3.1, we use textual inversions to replicate the ImageNet distribution. To evaluate the relevance of this approach, we generate 200 images of 100 randomly selected ImageNet classes using standard SD2.0 and SD2.0 with the textual inversions of IN*. To illustrate the distribution gap, we compute the accuracies for ResNet-50 (DeiT). They achieve an accuracy of 68.2% (71.6%) for the SD distribution and 74.1% (79.1%) for the IN* distribution, which equals accuracy drops of 6% (8%) for both classifiers. This is significantly closer to the performance on the original ImageNet distribution. We perform all the following experiments using the IN* distribution. We use SD2.0 and we activate the LoRA adapters with the selected scale for the last 75% of the noise steps.

Due to the computational complexity, we perform sliding for 100 classes. To get an estimate of the robustness on the full scale of ImageNet, we classify based on 1000 classes using off-the-shelf classifiers without applying classifier masking, as done by Hendrycks et al. (2021a). We ablate how the number of classes influences the robustness evaluations in Appendix A.5.2.

The selection of the shifts is mainly inspired by ImageNet-R Hendrycks et al. (2021a) (8 shifts) and the OOD-CV dataset Zhao et al. (2022) (6 shifts) to consider a diverse set of nuisance shifts that modulate the appearance and style or the background and occlusion. Specifically, we consider the following 14 shifts: cartoon style, plush toy style, pencil sketch style, painting style, design of sculpture, graffiti style, video game renditions style, style of a tattoo, heavy snow, heavy rain, heavy fog, heavy smog, heavy dust, and heavy sandstorm.

**Filtering details.** Our OOC filtering mechanism reaches a TPR of 87.9% and an FPR of 12.0% with an accuracy of 88.0%, while the naive CLIP-based thresholding reaches a TPR of 89.9% and an FPR of 35.7% with an accuracy of 65.1%. We plot the classification accuracy of DINOv2-R and ResNet-50 for the labeled and the filtered versions in Fig. 5. We observe comparable accuracy drops on both the manually-labeled and the filtered datasets. To further support the realism of our generated

Table 1: **rCE along the model axes.** We choose the average relative corruption error Hendrycks & Dietterich (2018) as a single metric to measure the performance of a model on our benchmark (lower is better). We provide results for all models in Table 2.

| Architecture | | Size | | Pre-Training | |
|---|---|---|---|---|---|
| ConvNext | 0.686 | DeiT3-S | 0.747 | DINOv1-IN1k | 0.636 |
| DeiT3 | 0.610 | DeiT3-M | 0.758 | MAE-IN1k | 0.732 |
| DeiT | 0.746 | DeiT3-B | 0.610 | MoCov3-IN1k | 0.669 |
| RN152 | 0.790 | DeiT3-L | 0.574 | SUP-IN1k | 0.926 |
| ViT | 0.926 | DeiT3-H | 0.583 | SUP-IN21k-1k | 0.722 |

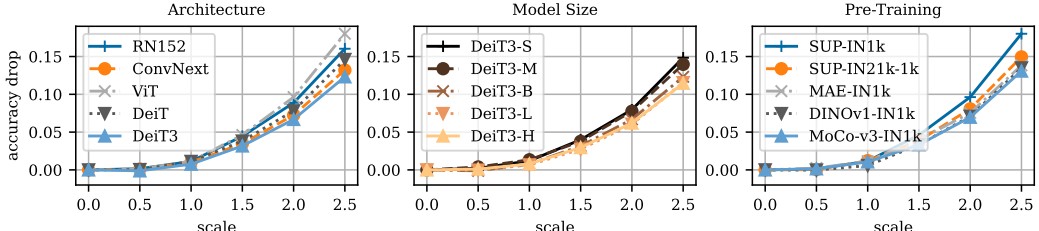

(a) Accuracy drops averaged over the whole benchmark. Architecture (*left*): We show models with the same training data and similar parameter counts. The selection of the architecture influences the accuracy drop. Model size (*center*): We show DeiT3 with various numbers of parameters. Increasing the model capacity results in lower accuracy drops. Pre-training paradigm and data (*right*): We show different pre-training paradigms: supervised, self-supervised (MAE, DINO, MoCo), and more data (IN21k), all using ViT-B/16. We present results for all shifts in Fig. 9.

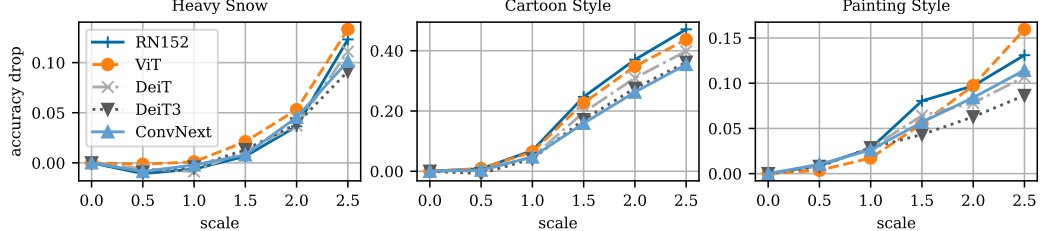

(b) Accuracy drops for three selected shifts. Models exhibit varying performance changes depending on the considered shifts. For snow and painting shifts, the ranking of the models changes. In contrast, the cartoon style shift results in a consistent model ranking. However, the OOD performance on cartoon-shifted images is drastically worse than the other shifts.

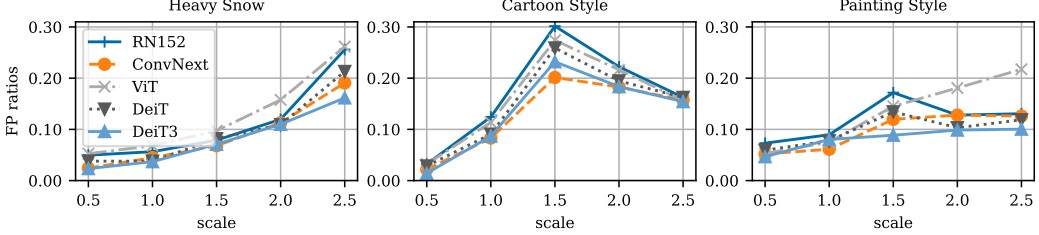

(c) Ratio of failure points per scale for various models and shifts. The distribution allows inferring at which scales various models fail most often. Different models fail at varying stages depending on the considered shifts. While the number of failure points gradually increases for the snow shift, most failure points occur around scale 1.5 for the cartoon style shift. We present results for all shifts in Fig. 10.

Figure 6: **Evaluation of accuracy drops and failure points.** We plot the averaged accuracy drops and failure points of selected models and provide the results of all evaluated models in Appendix A.2.

images, we fine-tune ResNet-50 with our data and show more than 10% gains on ImageNet-R (see Appendix A.3).

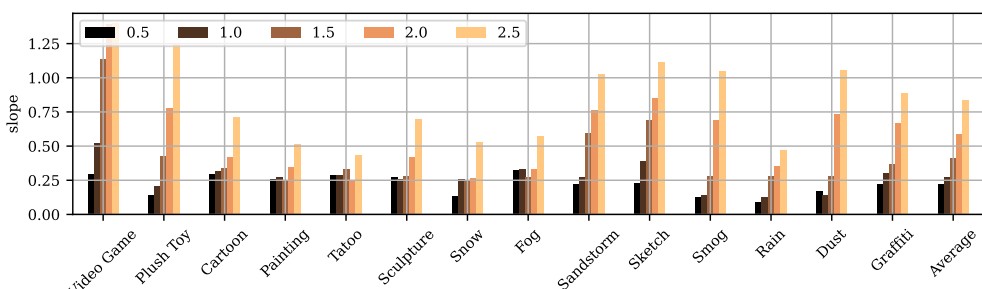

Figure 7: **Relation between ID and OOD accuracy.** We report the slope of the linear fit between ID and OOD accuracy using 16 supervised ImageNet-trained models for all evaluated shifts. The relation varies for different shifts and scales between 0.5 and 2.5.

## 4.3 ANALYSIS AND FINDINGS

In this subsection, we discuss the main findings on our benchmark. Following Hendrycks et al. (2021a), we report the accuracy drops for 5 scales averaged over 14 diverse shifts as a measure of robustness in Fig. 6a. Table 1 compares models using the average relative corruption errors as proposed by Hendrycks & Dietterich (2018). We also provide results for three exemplary shifts in Fig. 6b. In addition, we report the distribution of failure points in Fig. 6c. We provide more evaluations in Appendix A.2.

**Considering multiple scales of a shift allows a more nuanced analysis of OOD robustness.** We present the accuracy drops for multiple scales and classifiers along the architecture axis in Fig. 6b. The results indicate that the model rankings measured by the accuracy drop change for different scales and shifts. For example, while the rankings remain consistent for the cartoon style (*right*) for all scales, the model rankings change significantly for the painting style shift: Here, ViT outperforms the other models on a lower scale but performs worse on large shift scales. Varying rankings also occur for other shifts (see Fig. 9 in the supplementary). To validate the observation of changed model rankings, we also evaluate multiple corruption levels of an examplary ImageNet-C corruption and show the results in Fig. 23 in the supplementary. We conclude from this observation that the average accuracy drop and the accuracy drops at specific nuisance scales do not always indicate the same model behavior, which provides experimental evidence for the need for a multi-scale robustness benchmarking dataset and adequate metrics.

**Model failure points differ across different types of shifts.** A failure point captures at which scale a model fails for the first time. Comparing the failure point distribution of various models largely differs for different shift types, as shown in Fig. 6c. We provide more results in Fig. 10 in the supplementary. Weather shifts, such as snow, typically correspond to slight appearance changes and mainly add a disturbance factor or occlusions to the image. Therefore, the failure rate increases gradually compared to some style shifts, for which models tend to fail more abruptly at a specific scale, as, *e.g.*, for the cartoon style at scale $s = 1.5$. An exemplary explanation for the abrupt shift for the cartoon shift might be the wrong classification of a class as the ImageNet class *comic book*.

**The relation between ID and OOD accuracy depends on the considered nuisance factor and its scale.** Miller et al. (2021) formalize the positive correlation between ID and OOD accuracy—classifiers tend to have a better OOD accuracy if they perform better on the training data ("*Accuracy-on-the-line*" phenomenon). To analyze the linear relation between ID and OOD accuracy for our benchmark, we compute the slope of the linear fit between ID and OOD accuracies of 16 ImageNet-trained models. Miller et al. (2021) have already shown that the slope varies for different datasets. In Fig. 7, we further observe that not only the considered shift but also its severity influence the slope of the linear fit. Refer to Appendix A.2.3 for the test statistics. We believe using our benchmark to investigate this relation more extensively is an interesting direction for future work.

**Transformers with modern training recipes outperform modern CNNs across all shift severities.** We present the average accuracy drops of various models with the same training data and a comparable number of parameters in Fig. 6a (*left*). DeiT3 consistently achieves the highest robustness on our benchmark, increasing the gap towards DeiT and ViT for stronger shifts. Interestingly, ResNet-152 is more robust than the standard ViT variant, but ConvNeXt outperforms the ResNet-

152 architecture. A modern CNN (ConvNext) outperforms vision transformers (ViT,DeiT) of the same size but it is less robust than a transformer with modern training recipes (DeiT3), despite having a higher ID accuracy. This observation is in line with the performance on ImageNet-R. However, our benchmark shows that the gap between ConvNext and DeiT3 does not increase for stronger shifts. We can observe that this behavior is not consistent for all shifts. Consider, *e.g.*, the failure point distribution in Fig. 6c (*Painting Style*), where DeiT3 has a gradually increasing failure point rate, while ConvNext depicts a sharp increase for scale $s = 1.5$.

**Self-supervised pre-training improves the OOD robustness.** To study the impact of the pre-training paradigm, we compare different learning objectives with the same ViT-B backbone and the same training data in Fig. 6a (*right*). We consider both the supervised and self-supervised (MAE, DINOv1, and MoCov3) paradigms. Using a self-supervised objective for pre-training followed by a fine-tuning protocol results in a better robustness for the same training data and model size. Considering the rCE metric in Table 1, the fine-tuned DINOv1 model achieves the best performance.

**Diffusion classifiers are less robust than discriminative models.** In addition, we also compare the robustness of an ImageNet-trained diffusion classifier (Li et al., 2023b) on our benchmark. Due to the heavy computational cost, we evaluate the accuracy drop of the DiT-based diffusion classifier for 1k images on a subset of our dataset (around 12k images) for the snow and the cartoon style shift. We apply the L1 loss computation strategy as proposed by Li et al. (2023b) since it results in the best performance. We compute the average accuracy drops as 0.106 / 0.07 / 0.05 for DiT / supervised ViT / MAE. Comparing on the smaller dataset with discriminative models, the diffusion classifier demonstrates a lower robustness on the evaluated shifts than the compared discriminative models despite having substantially more parameters. The gap is increasing for larger severity levels.

**More training data improves the robustness.** In Fig. 6a (*right*), we observe that more training data benefits OOD robustness for all scales. For example, compared with the supervised model trained on IN1k, pre-training on IN21k positively impacts the OOD robustness for all scales. This might be explained by the fact that the tested distribution is less OOD for the model (Miller et al., 2021).

In summary, we show that benchmarking with generative continuous shifts allows systematically studying the model robustness via easily scalable synthetic data. Our study underscores that considering multiple-scale nuisance shifts provides a more nuanced view of the model robustness, as the performance drops can vary across different nuisance shifts and scales. Besides, the relation between ID and OOD accuracy not only depends on the considered nuisance factor but also on its severity. Therefore, instead of aggregating the robustness evaluation into a single metric, we motivate the community to report the accuracy with different shift scales and the failure points for a more comprehensive understanding of model robustness.

## 5 CONCLUSION

The key advantage of using generative models for benchmarking is the ability to perform diverse nuisance shifts in a controlled and scalable way. This work filled a gap in generative benchmarking by introducing CNS-Bench, an evaluation method that performs diverse, realistic, fine-grained, and continuous nuisance shifts at multiple scales. We further added a new dimension for benchmarking robustness by introducing the concept of failure points. Our systematic evaluation of classifiers revealed new insights along three axes (architecture, number of parameters, pre-training paradigm and data) and demonstrated the importance of continuous shifts in assessing the model robustness. Furthermore, we studied the necessity of removing out-of-class samples when benchmarking with diffusion-generated images. **Limitations and Future Work.** While our approach allows for diverse continuous nuisance shifts, it does not eliminate all confounders, meaning failures cannot always be solely attributed to the targeted nuisance concept. This highlights an inherent challenge for generative benchmarking approaches, and future advances in generative models could help mitigate these confounding factors. Additionally, while we have carefully addressed this issue in our work, we acknowledge that using generated images can lead to biases arising from the real vs. synthetic distribution shift.

We hope this benchmark can encourage the community to continue working on more high-quality generative benchmarks and to adopt generated images as an additional source for systematically evaluating the robustness of vision models.

# 6 REPRODUCIBILITY STATEMENT

All steps of our benchmarking pipeline are reproducible: We provide our datasets and implementation as part of the supplementary material, which includes code to reproduce training of LoRA adapters, generation of images, filtering, and evaluation of all classifiers. We also include all evaluated classification results for all images of the dataset in the shared code. All classifiers are evaluated in a standardized way using the *easyrobust* (Mao et al., 2022) framework.

The supplementary material contains more details about the implementation, the computation of metrics, the labeling, and the filtering strategies.

We also refer to our datasheet in Appendix B.

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

# A  APPENDIX

This appendix provides supplementary information that is not elaborated in our main paper: We will discuss more details about the benchmarking dataset, the filtering, and image generation strategy. Additionally, we will provide more results. We refer to GoogleDrive [1] folder for the benchmarking dataset, the filtering dataset, and the code to reproduce training, generation, filtering, and benchmarking.

## A.1  BENCHMARK DETAILS

This section provides more details about the benchmarking dataset. We first discuss the presented metrics. We then provide examples of the dataset and its distribution.

### A.1.1  ACCESS TO BENCHMARKING DATASET

The dataset contains $192, 168$ images in total, with $32, 028$ images per scale. We share all images on Google Drive in the folder *CNS_dataset*. Additionally, we add the anonymized metadata, including the annotations, as a JSON file. We will publish the data on our own servers upon submission. Currently, data is accessible via Google Drive and can be downloaded by running the following commands:

```
# Install gdown package
pip install gdown

# Download image files from Google Drive (22GB)
gdown https://drive.google.com/uc?id=1GYQb1dHu26mcklnMHtiySjZpigu-CmqN
# Download annotations from Google Drive
gdown https://drive.google.com/uc?id=1q4aS6oCdZx3jry6y92i3ZgoyNmHXZeN9

# Unzip the downloaded file
unzip benchmark.zip
```

### A.1.2  LIST OF SHIFTS AND EXAMPLE IMAGES

The results are averaged over the following 14 shifts: cartoon style, plush toy style, pencil sketch style, painting style, design of sculpture, graffiti style, video game renditions style, style of a tattoo, heavy snow, heavy rain, heavy fog, heavy smog, heavy dust, and heavy sandstorm (see examples in Fig. 33 and Fig. 34). We train the sliders using the prompt template "A picture of a {class} in {shift}".

## A.2  MORE RESULTS

### A.2.1  LABELED DATASET

Fig. 8 presents more classifier results on the labeled dataset.

### A.2.2  LARGE BENCHMARK

We provide a table of accuracies and accuracy drops for all evaluated models and scales and the average accuracy and accuracy drop in Table 3. As discussed in the main paper, we also provide the accuracy drops for the ResNet family in Fig. 12. Similar to the observations in Table 3, larger models result in a lower accuracy drop in average. Fig. 9 provides a more nuanced view on the model performances accross various architectures on all shifts. We provide functionality to load the classification results for all images of the dataset in the shared code. All results are computed in a standardized way using the *easyrobust* (Mao et al., 2022) framework.

The accuracies for the diffusion classifier are depicted in Fig. 21. Similar to the discussion in the paper, the results showcase that the generative classifier is less robust than a supervised classifier.

---

[1] https://drive.google.com/drive/folders/1twcuMLBSvy_lIRYhssiwivBv9eKsA_ul?usp=sharing

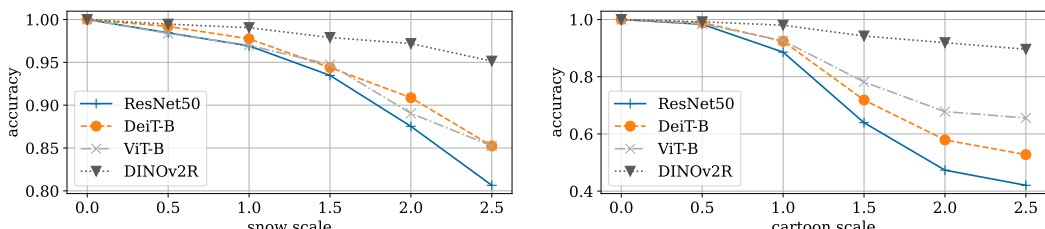

Figure 8: **Classification accuracy on the labeled dataset for snow and cartoon shifts.** The accuracy drops on the labeled dataset showcase that various classifiers have varying sensitivities on different shifts.

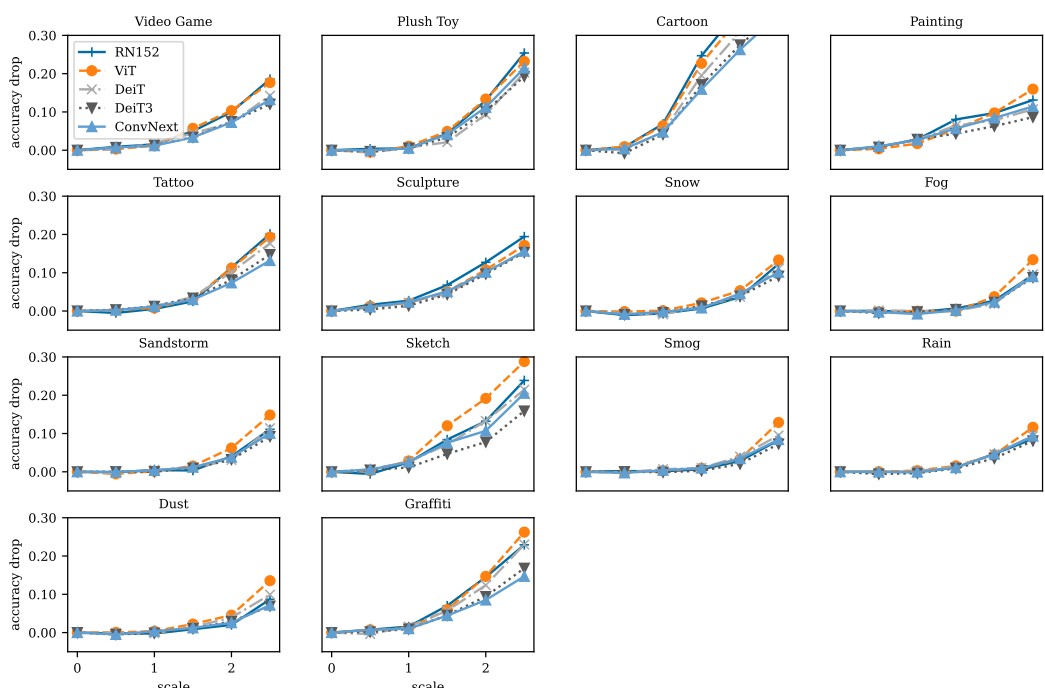

Figure 9: **Accuracy drops of various architectures for all shifts.** We present the accuracy drops for all shifts in our benchmark. The performance gaps vary for different shifts and scales.

We use the DiT-based diffusion classifier trained on ImageNet-1k using the available framework (Li et al., 2023a) and the default hyper-parameters with a resolution of 256. Due to high computational costs, we compute the results for 100 classes, four scales, for the snow and cartoon style shift, and for at most 20 seeds per class, scale, and shift.

### A.2.3 STATISTICS OF ACCURACY-ON-THE-LINE COMPUTATION

Fig. 17 provides the p-values of the linear regression corresponding to the presented results in Fig. 7.

### A.3 FINE-TUNING WITH SYNTHETIC DATA

We fine-tune a ResNet-50 classifier using our synthetic data. We compare the original ImageNet-trained model to a model fine-tuned using 50% synthetic data and 50% ImageNet training data. As shown in Table 5, the fine-tuned model leads to improved performance on the shifted real-world dataset, without a significant decline on the original ImageNet dataset.

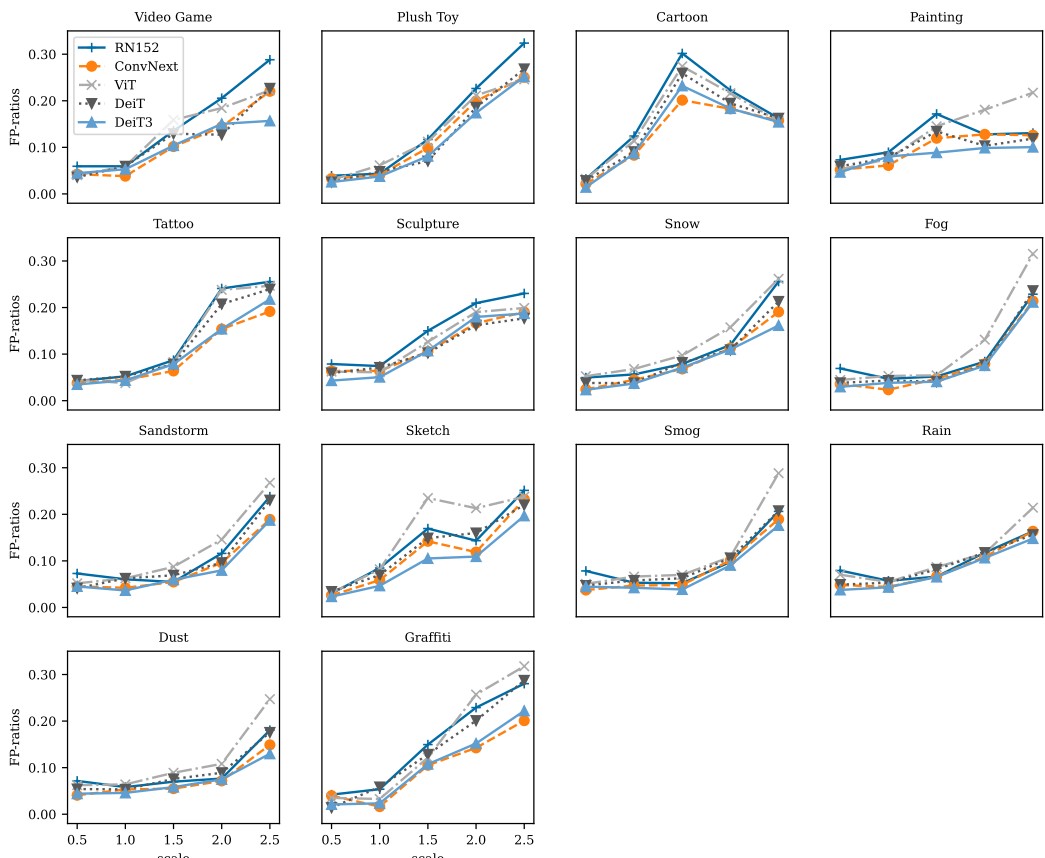

Figure 10: **Failure point distributions for all shifts.** We present the failure point distributions for all shifts in our benchmark. The failure point distributions vary for different shifts, quantifying the different ways the shifts influence model performance.

### A.4 ACCURACY DROPS ON IMAGENET-C

To provide more evidence that the model rankings change for different scales, we consider 7 levels of contrast as a deterministic example corruption from ImageNet-C, based on the implementation of Hendrycks & Dietterich (2018). We present the accuracy drops for all corruption levels in Fig. 23. A global averaged metric fails to capture such variations.

### A.5 IMPLEMENTATION DETAILS

In this section, we provide more implementation details about the dataset generation process.

#### A.5.1 IMPLEMENTATION DETAILS FOR IMAGE GENERATION

We use the standard diffusers (von Platen et al., 2022) pipeline for Stable Diffusion 2.0, the DDIM (Song et al., 2021) sampler with 100 steps and a guidance scale of 7.5, seeds ranging from 1 to 50.

#### A.5.2 ABLATION OF IMAGE GENERATION

We ablate how the number of classes influences the robustness evaluations in Fig. 25. For a more efficient computation, we use the `UniPCMultistepScheduler` sampler with 20 steps (Zhao et al., 2023). In addition to 100 sliders for 14 shifts, we also publish the sliders for all 1000 ImageNet classes for the shifts snow and cartoon.

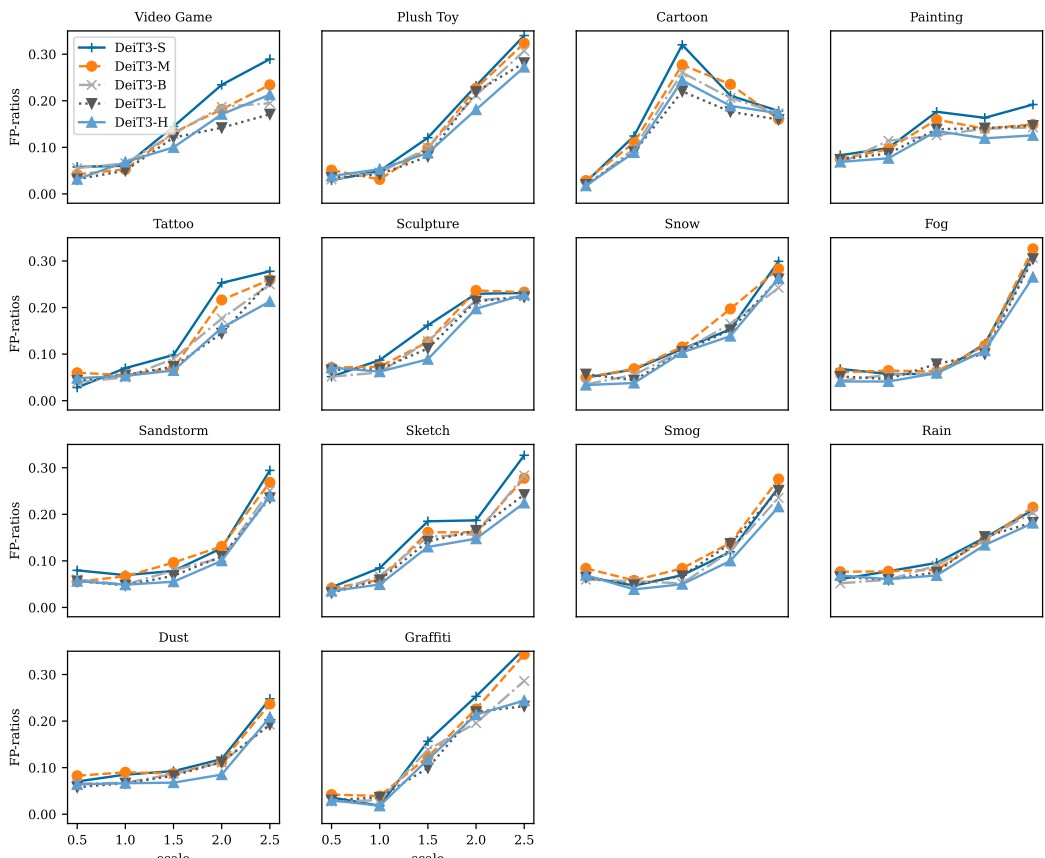

Figure 11: **Failure point distributions for all shifts.** We present the failure point distributions for all shifts in our benchmark along the model size axis of DeiT3. The failure point distributions vary for different shifts.

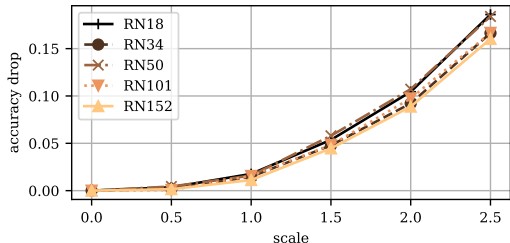

Figure 12: **Robustness evaluation for ResNet model family.** We compute the accuracy drops for all scales when varying the model size for a set of ResNet models. Larger models result in a better OOD robustness.

### A.5.3 TEXT-BASED CONTINUOUS SHIFT

A naive approach for realizing continuous shifts involves computing the difference between two corresponding CLIP embeddings. We explored this strategy following the implementation of Baumann et al. (2024), but we did not achieve robust nuisance shifts for the variety of classes we considered and we present some examples in Fig. 26. We achieve reasonable results for some classes (*e.g.*, upper row). However, we observe the following issues arising from this strategy: (1) The semantic structures clearly change, which involves other factors of variation. This does not allow the computation of a failure point along one sliding trajectory. (2) depicted in middle row: For some classes,

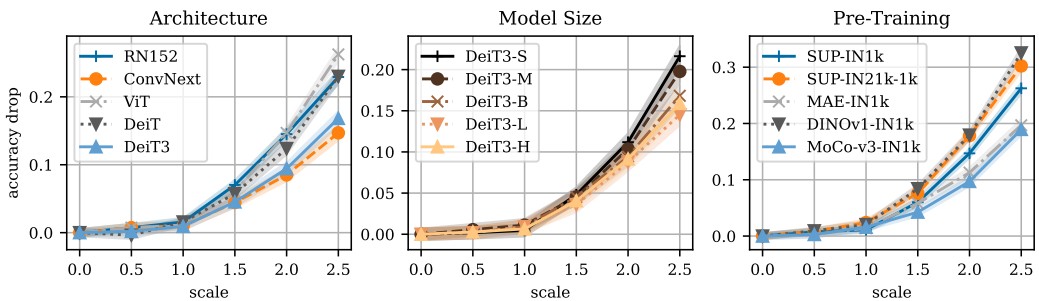

Figure 13: **Accuracy drops with confidence intervals.** The accuracy drops are depicted for the three evaluation axes averaged over all shifts including the confidence interval of the accuracy computation.

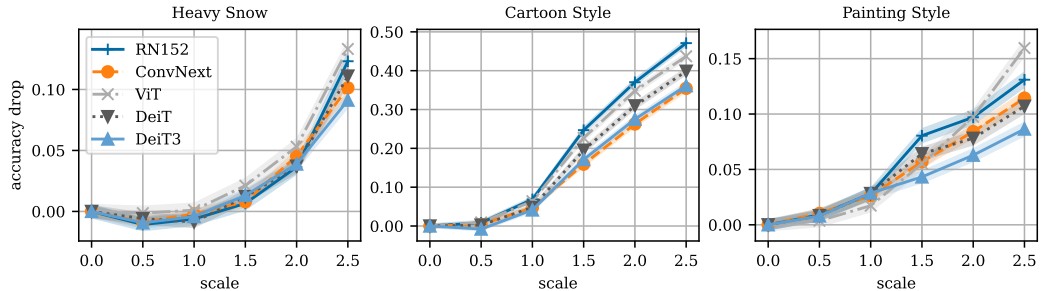

Figure 14: **Accuracy drops with confidence intervals.** The accuracy drops are depicted for the three shifts along the model axes including the one-sigma confidence interval of the accuracy computation. The results show that some ranking changes are statistically stable.

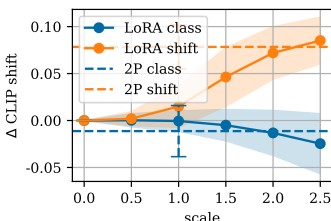

Figure 15: **Delta CLIP score with confidence.** The $\Delta$ CLIP score is plotted for various scales and averaged over all shifts including the standard deviation. The deviations are high. However, the can be attributed to the fact the range of the shift alignments varies for different shift types.

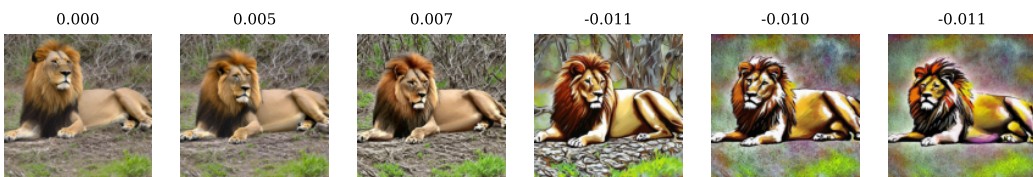

Figure 16: **Example of an incorrect CLIP alignment measure.** The CLIP alignment is applied as a measure to quantify the strength of the applied nuisance shift. However, this metric is not always correctly capturing the shift. Images with increasing slider scales and painting shift are represented. However, the alignment to the prompt "a picture in painting style" drops, as the $\Delta$ CLIP difference to the first image depicted as the image titles demonstrates.

Table 2: **mCE and mean rCE.** We present the mean corruption error and the mean relative corruption error for all evaluated models.

| Model | CE | rCE |
|---|---|---|
| alexnet | 1.000 | 1.000 |
| clip_resnet101 | 0.532 | 0.563 |
| clip_resnet50 | 0.715 | 0.587 |
| clip_vit_base_patch16_224 | 0.420 | 0.230 |
| clip_vit_base_patch32_224 | 0.487 | 0.591 |
| clip_vit_large_patch14_224 | 0.445 | 0.228 |
| clip_vit_large_patch14_336 | 0.419 | 0.274 |
| convnext_base.fb_in1k | 0.359 | 0.686 |
| convnext_large.fb_in1k | 0.354 | 0.672 |
| convnext_small.fb_in1k | 0.353 | 0.609 |
| convnext_tiny.fb_in1k | 0.393 | 0.809 |
| convnextv2_base.fcmae_ft_in1k | 0.322 | 0.680 |
| convnextv2_huge.fcmae_ft_in1k | 0.283 | 0.553 |
| convnextv2_large.fcmae_ft_in1k | 0.297 | 0.568 |
| deit3_base_patch16_224.fb_in1k | 0.396 | 0.610 |
| deit3_huge_patch14_224.fb_in1k | 0.353 | 0.583 |
| deit3_large_patch16_224.fb_in1k | 0.382 | 0.574 |
| deit3_medium_patch16_224.fb_in1k | 0.387 | 0.758 |
| deit3_small_patch16_224.fb_in1k | 0.400 | 0.747 |
| deit_base_patch16_224.fb_in1k | 0.437 | 0.746 |
| dino_vit_base_patch16 | 0.504 | 0.851 |
| dinov1_vit_base_patch16 | 0.381 | 0.636 |
| dinov2_vit_base_patch14 | 0.350 | 0.524 |
| dinov2_vit_base_patch14_reg | 0.311 | 0.456 |
| dinov2_vit_giant_patch14 | 0.321 | 0.431 |
| dinov2_vit_giant_patch14_reg | 0.311 | 0.426 |
| dinov2_vit_large_patch14 | 0.298 | 0.349 |
| dinov2_vit_large_patch14_reg | 0.296 | 0.370 |
| dinov2_vit_small_patch14 | 0.351 | 0.639 |
| dinov2_vit_small_patch14_reg | 0.330 | 0.627 |
| mae_vit_base_patch16 | 0.386 | 0.732 |
| mae_vit_huge_patch14 | 0.303 | 0.542 |
| mae_vit_large_patch16 | 0.328 | 0.571 |
| mocov3_vit_base_patch16 | 0.379 | 0.669 |
| resnet101.a1_in1k | 0.491 | 0.842 |
| resnet152.a1_in1k | 0.498 | 0.790 |
| resnet18.a1_in1k | 0.493 | 0.954 |
| resnet34.a1_in1k | 0.440 | 0.843 |
| resnet50.a1_in1k | 0.485 | 0.945 |
| vit_base_patch16_224.augreg_in1k | 0.569 | 0.926 |
| vit_base_patch16_224.augreg_in21k_ft_in1k | 0.460 | 0.722 |
| vit_base_patch16_clip_224.openai_ft_in1k | 0.282 | 0.482 |

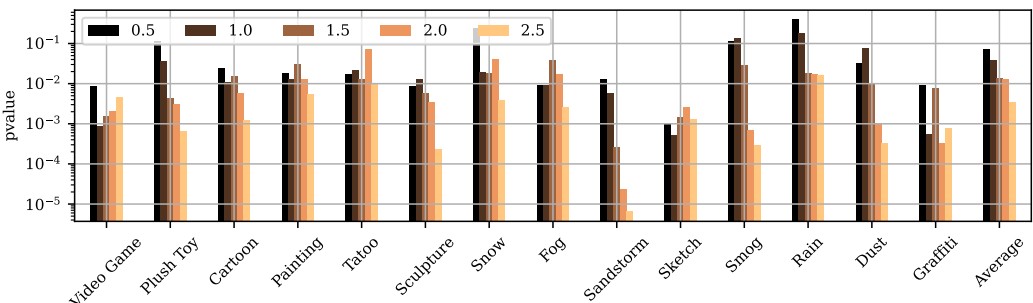

Figure 17: p-values of the linear regressions corresponding to the plot in Fig. 7: The p-value is smaller than $0.05$ for most scales and shifts, providing evidence for the statistical significance of our statements.

the naive approach is very unstable, resulting in OOD samples that do not represent realistic images. We did not reach significantly better results when applying a delayed sampling technique for the delta embedding. (3) depicted in the bottom row: Applying the delta in text-embedding space does not always result in a consistent increase of the considered shift.

### A.5.4 EVALUATION OF THE APPLIED SLIDER SHIFT

We evaluate whether our sliders always increase the shift, as measured by the $\Delta$ CLIP score. For this purpose, we compute the $\Delta$ CLIP score scores when increasing the slider scale by $0.5$. Here,

Table 3: **Accuracy evaluations.** We present the accuracies and accuracy drops of all evaluated classifiers.

| | Shift Scale | | | | | | | | | | | |
| | Accuracy | | | | | | | Accuracy Drop | | | | |
| model | 0 | 0.5 | 1 | 1.5 | 2 | 2.5 | avg | 1 | 1.5 | 2 | 2.5 | avg |
| clip_resnet50 | 0.81 | 0.81 | 0.8 | 0.78 | 0.74 | 0.67 | 0.77 | 0.01 | 0.03 | 0.07 | 0.14 | 0.04 |
| clip_resnet101 | 0.86 | 0.86 | 0.85 | 0.83 | 0.81 | 0.74 | 0.82 | 0.01 | 0.03 | 0.06 | 0.12 | 0.04 |
| clip_vit_base_patch16_224 | 0.87 | 0.88 | 0.88 | 0.87 | 0.86 | 0.81 | 0.86 | -0.00 | 0.01 | 0.02 | 0.06 | 0.02 |
| clip_vit_base_patch32_224 | 0.87 | 0.87 | 0.86 | 0.85 | 0.83 | 0.77 | 0.84 | 0.01 | 0.02 | 0.04 | 0.1 | 0.03 |
| clip_vit_large_patch14_224 | 0.87 | 0.87 | 0.87 | 0.86 | 0.85 | 0.82 | 0.86 | -0.00 | 0.01 | 0.02 | 0.05 | 0.01 |
| clip_vit_large_patch14_336 | 0.88 | 0.88 | 0.88 | 0.87 | 0.86 | 0.83 | 0.87 | 0.00 | 0.01 | 0.02 | 0.05 | 0.01 |
| convnext_tiny.fb_in1k | 0.92 | 0.92 | 0.91 | 0.88 | 0.84 | 0.77 | 0.87 | 0.01 | 0.04 | 0.08 | 0.15 | 0.05 |
| convnext_small.fb_in1k | 0.92 | 0.93 | 0.92 | 0.89 | 0.86 | 0.8 | 0.89 | 0.01 | 0.03 | 0.07 | 0.13 | 0.04 |
| convnext_base.fb_in1k | 0.93 | 0.93 | 0.92 | 0.89 | 0.85 | 0.79 | 0.89 | 0.01 | 0.03 | 0.07 | 0.13 | 0.04 |
| convnext_large.fb_in1k | 0.93 | 0.92 | 0.92 | 0.89 | 0.86 | 0.8 | 0.89 | 0.01 | 0.04 | 0.07 | 0.12 | 0.04 |
| convnextv2_base.fcmae_ft_in1k | 0.93 | 0.93 | 0.92 | 0.9 | 0.87 | 0.82 | 0.9 | 0.01 | 0.04 | 0.07 | 0.12 | 0.04 |
| convnextv2_large.fcmae_ft_in1k | 0.94 | 0.93 | 0.93 | 0.91 | 0.88 | 0.84 | 0.91 | 0.01 | 0.03 | 0.05 | 0.1 | 0.03 |
| convnextv2_huge.fcmae_ft_in1k | 0.94 | 0.93 | 0.93 | 0.91 | 0.89 | 0.84 | 0.91 | 0.01 | 0.03 | 0.05 | 0.09 | 0.03 |
| deit3_small_patch16_224.fb_in1k | 0.92 | 0.92 | 0.91 | 0.88 | 0.84 | 0.77 | 0.87 | 0.01 | 0.04 | 0.08 | 0.15 | 0.05 |
| deit3_base_patch16_224.fb_in1k | 0.91 | 0.91 | 0.9 | 0.88 | 0.84 | 0.79 | 0.87 | 0.01 | 0.03 | 0.07 | 0.12 | 0.04 |
| deit3_medium_patch16_224.fb_in1k | 0.92 | 0.92 | 0.91 | 0.88 | 0.84 | 0.78 | 0.88 | 0.01 | 0.04 | 0.08 | 0.14 | 0.05 |
| deit3_large_patch16_224.fb_in1k | 0.91 | 0.91 | 0.9 | 0.88 | 0.85 | 0.8 | 0.88 | 0.01 | 0.03 | 0.06 | 0.12 | 0.04 |
| deit3_huge_patch14_224.fb_in1k | 0.92 | 0.92 | 0.91 | 0.89 | 0.86 | 0.81 | 0.89 | 0.01 | 0.03 | 0.06 | 0.11 | 0.04 |
| deit_base_patch16_224.fb_in1k | 0.9 | 0.9 | 0.89 | 0.87 | 0.83 | 0.76 | 0.86 | 0.01 | 0.04 | 0.08 | 0.15 | 0.05 |
| dino_lp_vit_base_patch16 | 0.9 | 0.9 | 0.89 | 0.85 | 0.8 | 0.71 | 0.84 | 0.01 | 0.05 | 0.1 | 0.19 | 0.06 |
| dinov1_ft_vit_base_patch16 | 0.91 | 0.91 | 0.90 | 0.88 | 0.84 | 0.84 | 0.87 | 0.01 | 0.03 | 0.07 | 0.04 | asd |
| dinov2_vit_small_patch14 | 0.92 | 0.92 | 0.91 | 0.89 | 0.86 | 0.81 | 0.89 | 0.01 | 0.03 | 0.06 | 0.11 | 0.04 |
| dinov2_vit_small_patch14_reg | 0.93 | 0.93 | 0.92 | 0.9 | 0.87 | 0.81 | 0.89 | 0.01 | 0.03 | 0.06 | 0.11 | 0.04 |
| dinov2_vit_base_patch14 | 0.91 | 0.91 | 0.91 | 0.89 | 0.87 | 0.82 | 0.89 | 0.00 | 0.02 | 0.04 | 0.09 | 0.02 |
| dinov2_vit_base_patch14_reg | 0.92 | 0.92 | 0.92 | 0.9 | 0.88 | 0.84 | 0.9 | 0.00 | 0.02 | 0.04 | 0.08 | 0.02 |
| dinov2_vit_large_patch14 | 0.92 | 0.92 | 0.92 | 0.91 | 0.89 | 0.86 | 0.9 | 0.00 | 0.01 | 0.03 | 0.06 | 0.02 |
| dinov2_vit_large_patch14_reg | 0.92 | 0.92 | 0.91 | 0.91 | 0.89 | 0.86 | 0.9 | 0.00 | 0.01 | 0.03 | 0.06 | 0.02 |
| dinov2_vit_giant_patch14 | 0.91 | 0.91 | 0.91 | 0.9 | 0.88 | 0.84 | 0.89 | 0.00 | 0.01 | 0.04 | 0.07 | 0.02 |
| dinov2_vit_giant_patch14_reg | 0.92 | 0.92 | 0.91 | 0.9 | 0.88 | 0.85 | 0.9 | 0.00 | 0.01 | 0.03 | 0.07 | 0.02 |
| mae_vit_base_patch16 | 0.92 | 0.92 | 0.91 | 0.88 | 0.84 | 0.78 | 0.88 | 0.01 | 0.04 | 0.08 | 0.14 | 0.05 |
| mae_vit_huge_patch14 | 0.93 | 0.93 | 0.92 | 0.9 | 0.88 | 0.84 | 0.9 | 0.01 | 0.03 | 0.05 | 0.1 | 0.03 |
| mae_vit_large_patch16 | 0.93 | 0.92 | 0.92 | 0.9 | 0.87 | 0.83 | 0.9 | 0.01 | 0.03 | 0.05 | 0.1 | 0.03 |
| mocov3_vit_base_patch16 | 0.92 | 0.92 | 0.91 | 0.88 | 0.85 | 0.79 | 0.88 | 0.01 | 0.03 | 0.07 | 0.13 | 0.04 |
| resnet18.a1_in1k | 0.9 | 0.9 | 0.88 | 0.85 | 0.8 | 0.72 | 0.84 | 0.02 | 0.05 | 0.1 | 0.19 | 0.06 |
| resnet34.a1_in1k | 0.91 | 0.91 | 0.9 | 0.86 | 0.82 | 0.75 | 0.86 | 0.01 | 0.05 | 0.09 | 0.17 | 0.05 |
| resnet50.a1_in1k | 0.91 | 0.9 | 0.89 | 0.85 | 0.8 | 0.72 | 0.85 | 0.02 | 0.06 | 0.11 | 0.18 | 0.06 |
| resnet101.a1_in1k | 0.9 | 0.9 | 0.88 | 0.85 | 0.8 | 0.73 | 0.84 | 0.02 | 0.05 | 0.1 | 0.17 | 0.06 |
| resnet152.a1_in1k | 0.89 | 0.89 | 0.88 | 0.85 | 0.8 | 0.73 | 0.84 | 0.01 | 0.04 | 0.09 | 0.16 | 0.05 |
| vit_base_patch16_224.augreg_in1k | 0.87 | 0.87 | 0.86 | 0.82 | 0.77 | 0.69 | 0.81 | 0.01 | 0.05 | 0.1 | 0.18 | 0.06 |
| vit_base_patch16_224.augreg_in21k_ft_in1k | 0.9 | 0.9 | 0.89 | 0.86 | 0.82 | 0.75 | 0.85 | 0.01 | 0.04 | 0.08 | 0.15 | 0.05 |
| vit_base_patch16_clip_224.openai_ft_in1k | 0.93 | 0.93 | 0.92 | 0.91 | 0.89 | 0.86 | 0.91 | 0.01 | 0.02 | 0.04 | 0.08 | 0.03 |

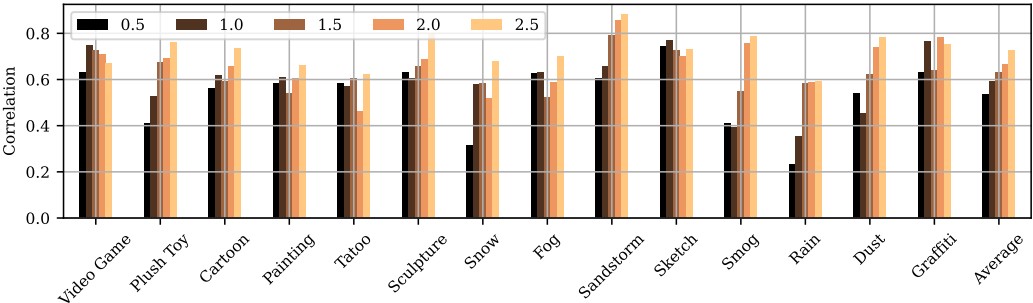

Figure 18: We report the linear correlation coefficients between ID and OOD accuracy using 16 supervised ImageNet-trained models for all evaluated shifts. The relation varies for different shifts and scales between 0.5 and 2.5.

the CLIP shift alignment increases for 73% of all cases for scales $s > 0$ and averaged over all shifts, demonstrating that increasing the slider weight results in a stronger severity of the desired shift.

### A.5.5 IMPLEMENTATION DETAILS FOR BENCHMARKING

We provide the code for training the LoRA adapters and for performing the sliding. For benchmarking all vision models, we integrate our new benchmark and additional models in the easyrobust (Mao et al., 2022) framework. We provide all classification results for all images of the dataset together with the code and the data in the supplementary material.

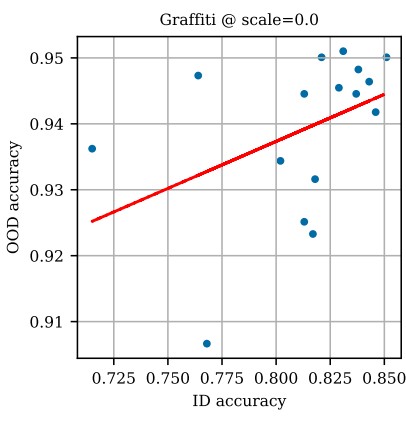 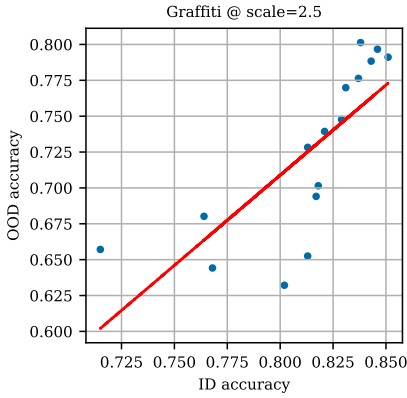

(a) Fit for no applied shift.     (b) Fit for a graffiti shift with scale 2.5.

Figure 19: **Linear fits of the ID and OOD accuracies.** We plot example linear fits of ID and OOD accuracies for the graffiti style. It can be observed that the slope increases for a larger scale.

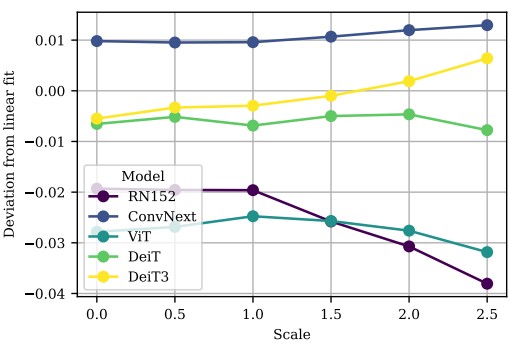

Figure 20: **Accuracy gains of models along the architecture axis.** We plot the accuracy gains averaged over all shifts after correcting for the effect of the ID-OOD accuracy slope. These gains are computed by substracting the effect of the linear fit (consider Fig. 19 for an example) from the OOD accuracies. After that correction, ConvNext performs better than DeiT3.

### A.5.6   DETAILS ABOUT THE USED COMPUTE

We used the internal cluster consisting of NVIDIA A40, A100, and RTX 8000 GPUs for running most of the experiments. Small-scale experiments are conducted on workstations equipped with RTX 3090. Training one LoRA adapter requires 1 to 2 hours depending on the used GPU, generating the images for one shift and class with 50 seeds and 6 scales requires 10 to 20 minutes. Thus, the training of the 1400 LoRA adapters took around 2000 GPU hours and the generation of the images around 350 GPU hours. Benchmarking all models using *easyrobust* required around 1000 GPU hours. The experiments to perform classification using the diffusion-classifier required around 4000 GPU hours.

## A.6   LABELING

In this section, we provide more details about the labeling dataset and strategy.

### A.6.1   DETAILS ON THE CREATION OF THE LABELED DATASET

To select a filter for detecting OOC samples, we collected a dataset for manual labeling: We pursue the following strategy: (i) In the first stage, 24k images are generated for 20 seeds, 5 LoRA scales, and 2 shifts per class for 100 random ImageNet classes in total. We select two very different shifts:

Table 4: **ImageNet validation accuracies and parameter count**. One the left, we plot model accuracies on the ImageNet validation dataset for all evaluated classifiers. On the right, we present the parameter counts for the used architectures.

| Model | IN/val |
|---|---|
| clip_resnet101 | 58.00 |
| clip_resnet50 | 55.00 |
| clip_vit_base_patch16_224 | 67.70 |
| clip_vit_base_patch32_224 | 62.60 |
| clip_vit_large_patch14_224 | 75.00 |
| clip_vit_large_patch14_336 | 76.30 |
| convnext_base.fb_in1k | 83.80 |
| convnext_large.fb_in1k | 84.30 |
| convnext_small.fb_in1k | 83.10 |
| convnext_tiny.fb_in1k | 82.10 |
| convnextv2_base.fcmae_ft_in1k | 84.90 |
| convnextv2_huge.fcmae_ft_in1k | 86.20 |
| convnextv2_large.fcmae_ft_in1k | 85.80 |
| deit3_base_patch16_224.fb_in1k | 83.70 |
| deit3_huge_patch14_224.fb_in1k | 85.10 |
| deit3_large_patch16_224.fb_in1k | 84.60 |
| deit3_medium_patch16_224.fb_in1k | 82.90 |
| deit3_small_patch16_224.fb_in1k | 81.30 |
| deit_base_patch16_224.fb_in1k | 81.80 |
| dino_lp_vit_base_patch16 | 78.10 |
| dino_v1_vit_base_patch16 | 82.49 |
| dinov2_vit_base_patch14 | 84.50 |
| dinov2_vit_base_patch14_reg | 84.60 |
| dinov2_vit_giant_patch14 | 86.60 |
| dinov2_vit_giant_patch14_reg | 87.10 |
| dinov2_vit_large_patch14 | 86.40 |
| dinov2_vit_large_patch14_reg | 86.70 |
| dinov2_vit_small_patch14 | 81.40 |
| dinov2_vit_small_patch14_reg | 80.90 |
| mae_vit_base_patch16 | 83.70 |
| mae_vit_huge_patch14 | 86.90 |
| mae_vit_large_patch16 | 86.00 |
| mocov3_vit_base_patch16 | 83.20 |
| resnet101.a1_in1k | 81.30 |
| resnet152.a1_in1k | 81.70 |
| resnet18.a1_in1k | 71.50 |
| resnet34.a1_in1k | 76.40 |
| resnet50.a1_in1k | 80.20 |
| vit_base_patch16_224.augreg_in1k | 76.80 |
| vit_base_patch16_224.augreg_in21k_ft_in1k | 77.70 |
| vit_base_patch16_clip_224.openai_ft_in1k | 85.20 |

| Model | Number of parameters (in million) |
|---|---|
| convnext_tiny | 29 |
| convnext_small | 50 |
| convnext_base | 89 |
| convnext_large | 198 |
| convnextv2_base | 89 |
| convnextv2_huge | 660 |
| convnextv2_large | 198 |
| deit3_small | 22 |
| deit3_medium | 39 |
| deit3_base | 87 |
| deit3_huge | 632 |
| deit3_large | 304 |
| deit_base | 87 |
| vit_base | 87 |
| vit_huge | 632 |
| vit_large | 307 |
| resnet18 | 12 |
| resnet34 | 22 |
| resnet50 | 26 |
| resnet101 | 45 |
| resnet152 | 60 |

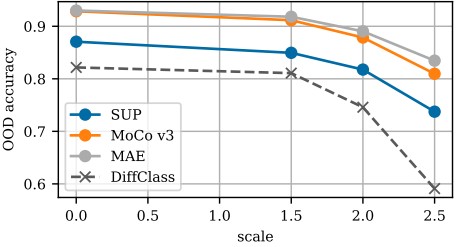

(a) Accuracies for heavy snow shift.

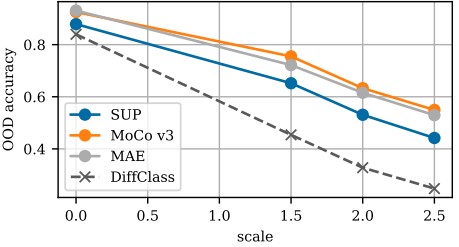

(b) Accuracies for cartoon style shift.

Figure 21: **Comparison of DiT classifier.** We report the OOD accuracies for two shifts for the DiT classifier (Li et al., 2023b) and discriminative classifiers. All models were trained on ImageNet-1k and are evaluated on the same subset of our benchmark. The diffusion classifier performs worse than the discriminative models.

One shift corresponds to a natural variation (snow), and the second shift corresponds to a style shift (cartoon style). (ii) We aim to find OOC samples that arise due to the application of the LoRA adapters. Therefore, we remove all images generated with a seed that results in a generated image that has a low CLIP text-alignment or is not classified classified correctly as the corresponding class even without the application of LoRA adapters. After removing such images, the labeling dataset consists of around 18k images. (iii) To reduce the labeling effort, we filter out all easy samples that (1) are correctly classified by DINOv2-ViT-L (Caron et al., 2021; Oquab et al., 2023) with a linear fine-tuned head and (2) one out of three classifiers (ResNet-50, DeiT-B/16, or ViT-B/16). (3) Additionally, the text alignment needs to be sufficiently high. (iv) Each hard image is labeled by

Table 5: **ImageNet-R performance after fine-tuning on our benchmark data.** ImageNet-R accuracy of the original ResNet-50 without fine-tuning and our model, fine-tuned on our benchmark.

| Evaluation Dataset | wo/ fine-tuning | w/ fine-tuning |
|---|---|---|
| IN/val | 80.15 | 78.11 |
| IN/R | 27.34 | 37.57 |

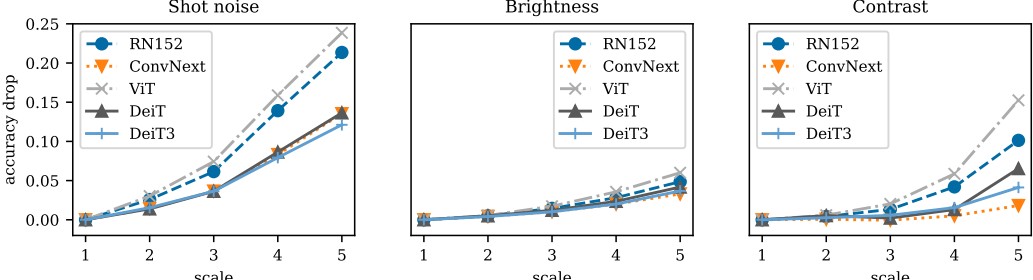

Figure 22: **Accuracy drops for three ImageNet-C corruptions and various architectures.** The model rankings change for different corruptions, underlining the importance of the selection of the corruption types or nuisance shifts for benchmarking the OOD robustness. Additionally, it can also be observed that the accuracy drops at varying rates for different shifts.

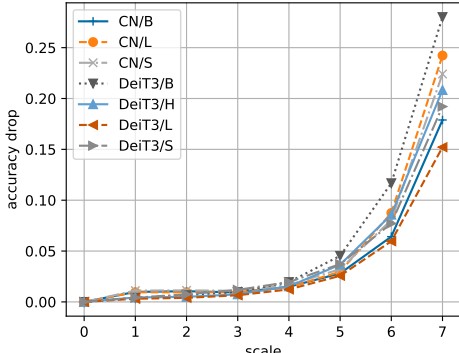

Figure 23: **Accuracy drops for contrast corruption.** We report the accuracy drops for seven severities of the contrast corruption, as defined in (Hendrycks & Dietterich, 2018). The model rankings change for different scales.

two human annotators. To increase the dataset quality, we include soft labels if the image partially includes some characteristics of the class. So, each annotator can choose from the labels 'class', 'partial class properties', and 'not class'. An image is defined as an out-of-class sample if at least one annotator considers the image as an OOC. For the remaining samples, an image is considered IC (in-class) if at least one annotator labeled the image a clear sample of the corresponding class

For the pre-filtering strategy (ii) and for the selection of easy samples (iii), we compute text-alignment using CLIP score and we remove all samples that have a CLIP similarity $s_{\text{CLIP-text-alignment}} > 24$, which approximately includes 90% of all ImageNet validation images (Vendrow et al., 2023). We use the implementation in *torchmetrics* with VIT-B/16. After removing the easy samples in step (iii), 2.7k images remain for labeling. We use the VIA annotation tool (Dutta & Zisserman, 2019; Dutta et al., 2016) to create the annotations. Each image is labeled by two humans. In total, 14 graduate students are involved in the labeling process. For all participants, we ensure sufficient motivation and they receive detailed instructions on how to perform the labeling

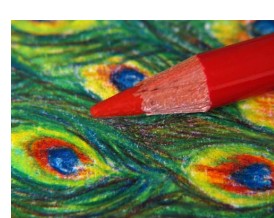 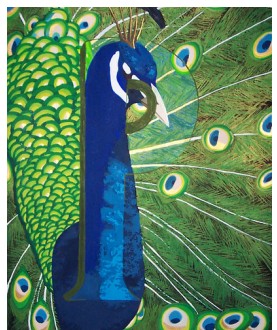 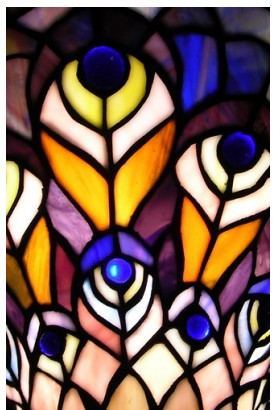

Figure 24: **ImageNet-R examples.** Example images of one class where the shape and perspective significantly change.

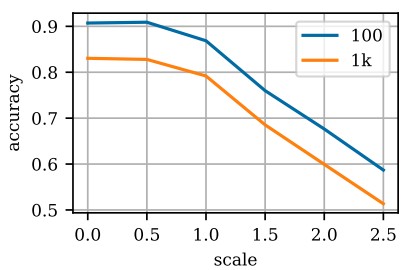

(a) Accuracy over various scales.

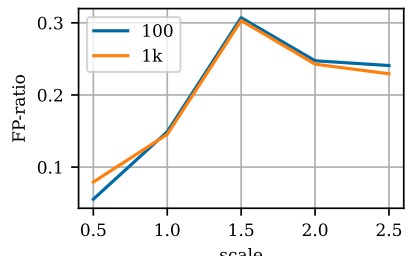

(b) Failure point distribution (normalized over the sum of failure points).

Figure 25: **Ablation of the number of ImageNet classes.** We compare the accuracies and failure points averaged over the selected 100 classes and all 1000 ImageNet classes for two shifts (snow and cartoon style). We report the results with ResNet-50. The results indicate that the initial accuracy estimate is overestimated but the accuracy drops averaged over the two shifts are in line.

(the full set of instructions is provided in Fig. 32). We provide the filtering statistics in Table 6. An example screenshot of the labeling tool is visualized in Fig. 27.

### A.6.2 LABELING DATASET

We provide the images for labeling in the provided URL as well. There, we include all images and metadata that allow inferring the class of each image and the tag, whether it is labeled automatically or by a human. The statistics of the labeling dataset are shown in Fig. 28.

### A.7 USER STUDY

We perform a user study on the final dataset using the same tooling as for the human labeling discussed in Appendix A.6 (iv). The user study includes 300 randomly sampled images and it is checked by two different individuals. In total, the user study involved seven people with different professions. 3 samples of our benchmark were considered as out-of-class samples, resulting in a ratio of 1% of failure cases with a margin of error of 0.5% for a one-sigma confidence level.

### A.8 APPLICATIONS OF TRAINED SLIDERS

We can combing various sliders by simply adding the corresponding LoRA adapters. We show an example application in Fig. 29.

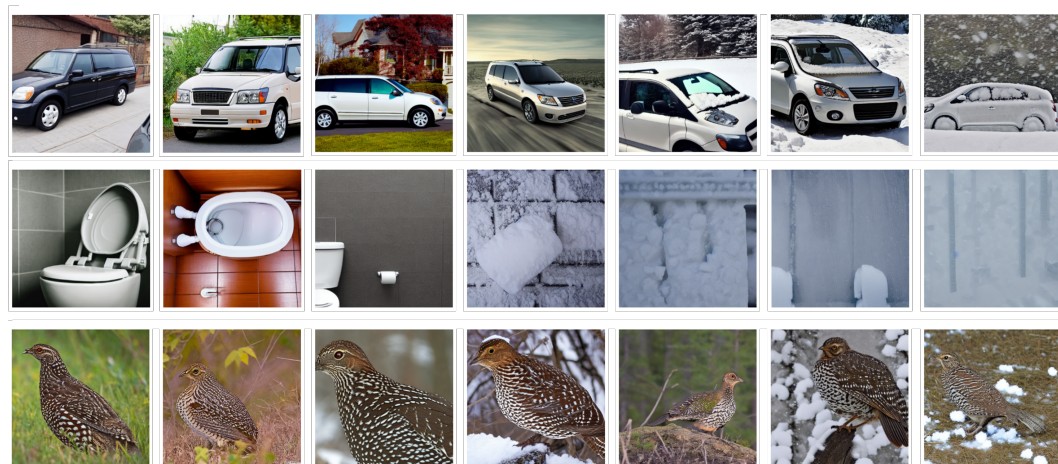

Figure 26: **Examples for text-based continuous shift.** The gradual increase can be successful. However, we observe that it fails for some classes (middle row) and is not consistently increasing (bottom row).

Table 6: **Statistics of filtering process.** We report the number of samples after various filtering stages. The stages are numbered according to the description in the main paper.

| Scale | Stage (i) | Stage (ii) | Stage (iii) | Stage (iv) |
|---|---|---|---|---|
| 0 | 4000 | 2966 | 2966 | 2966 |
| 0.5 | 4000 | 2966 | 2929 | 2955 |
| 1 | 4000 | 2966 | 2813 | 2906 |
| 1.5 | 4000 | 2966 | 2479 | 2740 |
| 2 | 4000 | 2966 | 2143 | 2498 |
| 2.5 | 4000 | 2966 | 1729 | 2110 |

## A.9 OOD-CV DETAILS

The Out-of-Distribution Benchmark for Robustness (OOD-CV) dataset includes real-world OOD examples of 10 object categories varying in terms of 5 nuisance factors: *pose*, *shape*, *context*, *texture*, and *weather*.

**Generation of images for synthetic OOD-CV** We generate the images for the synthetic OOD-CV dataset using a larger number of noise steps (85%) and more scale (between 0 and 3) since the classes occur more often in the dataset for training CLIP and Stable Diffusion. We use SD2.0 and not the dataset interfaces provided by Vendrow et al. (2023) since the class differences are less subtle and the samples of OOD-CV originate from two different datasets.

**Training subset** The OOD-CV benchmark provides a training subset of 8627 images. We train different state-of-the-art classifiers (i.e., ResNet-50 (He et al., 2016), ViT-B/16 (Dosovitskiy et al., 2020), and DINO-v2-ViT (Oquab et al., 2023)) for classification. We finetune each baseline during 50 epochs with an early stopping set to 5 epochs. In order to make baselines more robust, we apply standard data augmentation such as scale, rotation, and flipping during training. The training subset is composed of images originating from different datasets, notably ImageNet (Deng et al., 2009) and Pascal-VOC (Everingham et al., 2010). It is important to notice that the distribution of these two subsets is slightly different, with a higher data quality for the ImageNet subset and a lower quality for the latter subset (more noise, smaller objects, different image sizes). We visualize a few examples of the training data in Fig. 31.

**Test subset annotations** In the test subset provided in the benchmark dataset, only the coarse individual nuisance factors (*e.g.*, *weather*, *texture*) are provided. In our setup, we

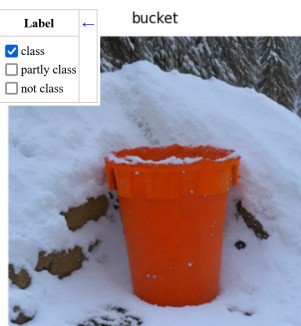

Figure 27: **Screenshot of labeling tool.** We plot a screenshot of an example image as it appeared during our labeling.

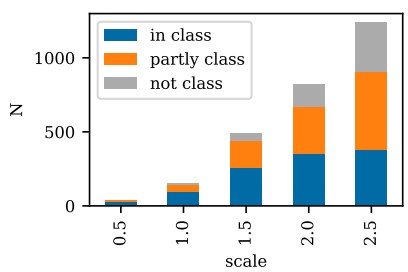

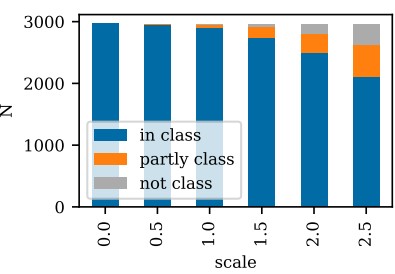

(a) For the human labeling dataset.

(b) For the complete filtering dataset.

Figure 28: **Statistics of labeling dataset.** We report the number of in-class, partially in-class, and out-of-class samples.

are interested in studying more fine-grained nuisance shifts, notably *rain*, *snow*, or *fog*. Hence, we had to assign some fine-grained annotation to all images containing *weather* nuisance shifts. Hence, we assign a fine-grained annotation by computing the CLIP similarity to the following texts: "`a picture of a {class} in {shift}`", where `class` is the ground truth class and `shift` the nuisance shift candidate *rain*, *snow*, or *fog* and "`a picture of a {class} without snow nor fog nor rain`". By applying a softmax on the similarity scores with the previous texts, we can assign the fine-grained nuisance shift *rain*, *snow*, *fog* or *unknown* for each image. We show more statistics in Table 7. By checking the results visually, we observe that all fine-grained nuisance shifts align with human perception and have a tendency towards classifying samples as *unknown* as soon as there is a small doubt. Note that by applying the same strategies to our generated data, we obtain an accuracy close to $100\%$.

**Nearest neighbor images of OOD-CV and CNS-Bench.** To illustrate the realism of our generated image, we compute the nearest neighbours using cosine similarity with CLIP image embedding and we plot it in Fig. 30.

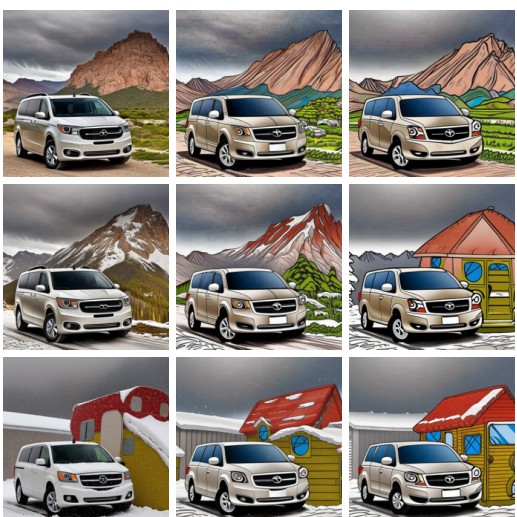

Figure 29: **Combination of Sliders.** We exemplarily show that sliders can be combined. Here, a snow slider (vertical axis) and a cartoon slider (horizontal axis) are linearly added for three scales.

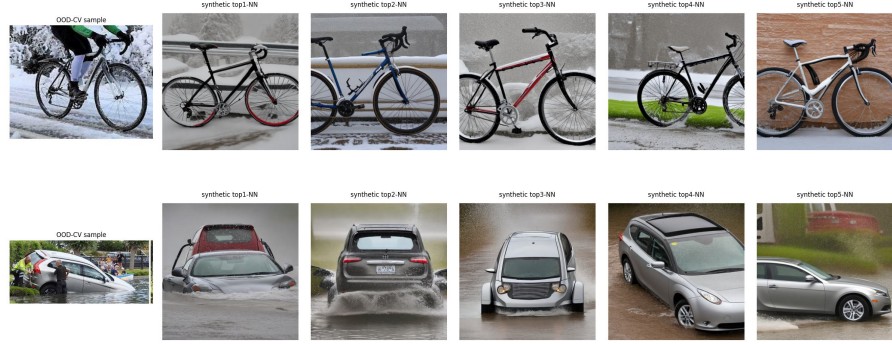

Figure 30: **Closest synthetic samples to two example OOD-CV images.** We find the top-5 nearest neighbours using cosine similarity with CLIP image embedding.

Table 7: **OOD-CV Statistics.** We report the number of images and accuracies for the weather subset.

| Shift | #images | Accuracy |
|---|---|---|
| Snow | 273 | 70.3 |
| Fog | 24 | 62.5 |
| Rain | 74 | 66.2 |
| Unknown | 129 | 66.7 |
| Total | 500 | 68.4 |

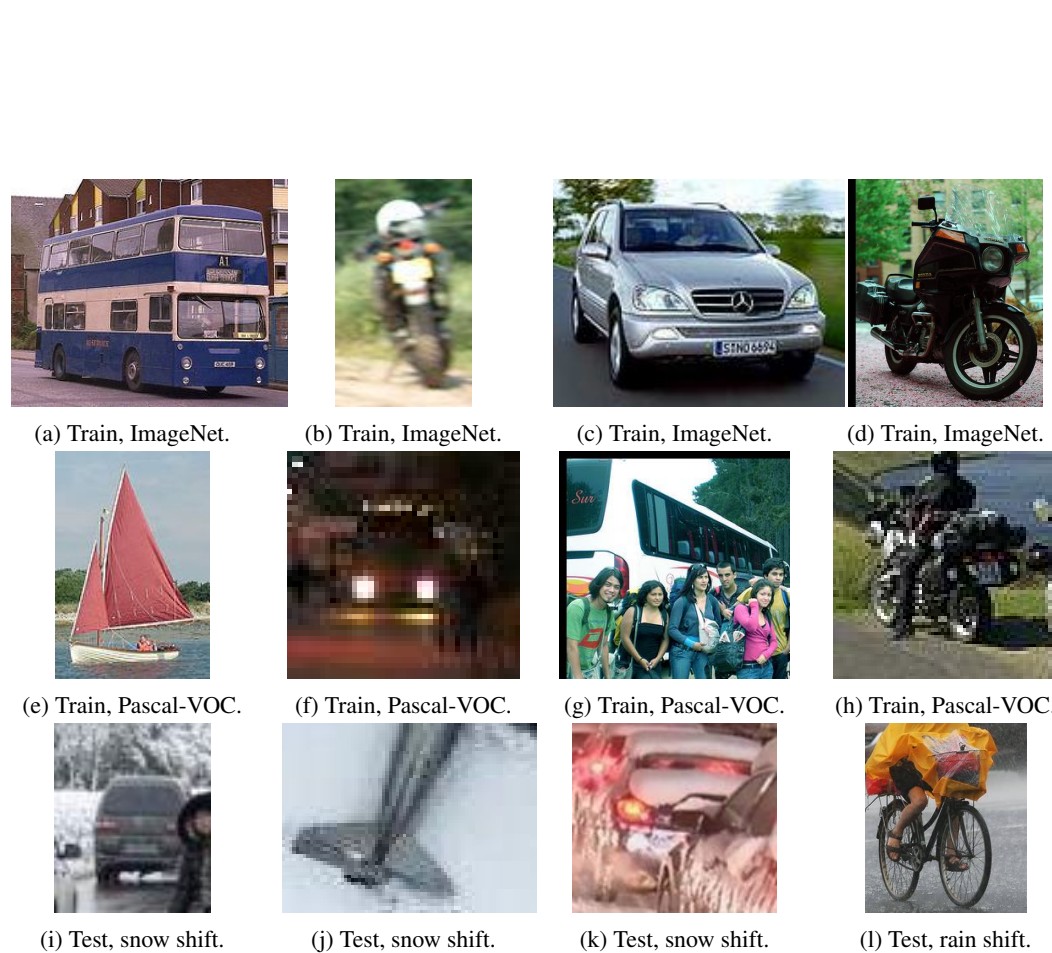

(a) Train, ImageNet.    (b) Train, ImageNet.    (c) Train, ImageNet.    (d) Train, ImageNet.

(e) Train, Pascal-VOC.    (f) Train, Pascal-VOC.    (g) Train, Pascal-VOC.    (h) Train, Pascal-VOC.

(i) Test, snow shift.    (j) Test, snow shift.    (k) Test, snow shift.    (l) Test, rain shift.

Figure 31: **OOD-CV example images.** We illustrate a set of example images from the training and the testing dataset of OOD-CV: (a-h) example from the training set, from ImageNet or Pascal-VOC. (i-l) Some examples for weather nuisance shifts. In the training set, we observe that images from the Pascal-VOC subset are usually of lower quality (*e.g.*, cropping, occlusion, resolution) compared to the ImageNet subset. In the test set, we see that that not fully disentangled (*e.g.*, (j) is only partially visible, (k) is partially occluded).

# Labeling task for out-of-class detection

**Motivation**: For benchmarking a classifier with synthetic images, we need to ensure that the generated images still correspond to the correct classes. To evaluate automatic filtering pipelines, we create a dataset with human labels. The dataset includes generated images with various levels of snow or cartoon style.

**Task:**
The goal is to detect images that do not belong to the corresponding ImageNet class (given as title).

Given an image, your task is to select one of three labels:
- ***class*:**
  - You can clearly recognize the class.
- ***partly class*:**
  - Given the class label, the class seems to correspond to the image.
  - You can recognize parts of the class but you are not very sure whether this is actually the class
  - You clearly see some characteristics of the class but it does not include all the important features.
- ***not class*:**
  - The considered image is clearly not the considered class.

The goal is to check whether the objects in the image correspond to a class or not. The goal is not to check whether the samples look realistic.

Every class starts with one realistic example image, taken from ImageNet. This image needs to be labeled as well. Since the example is just one illustrative example, not depicting the diversity of the class, it is recommended to use Google picture search to get an intuition of how the object looks in case one is not familiar with the class.
Some of the consecutive class samples will be similar. They are generated with the same seed but with varying snow or cartoon levels.

Some examples for class, partly class, and not class:

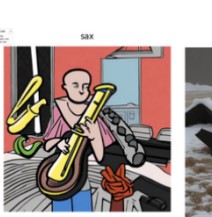

1) ***class*:** This animal can be clearly described as a fox at first glance. Also, the bucket can be easily recognized.

2) ***partly class*:** The shape and size seems to fit a ladybug. However, the black dots are missing. The other picture might be a cartoon-like illustration of apples. However, this can be argued. It is not clear.

3) ***not class*:** First example: This is supposed to be a sax but it is clearly not recognizable as a sax. Second example: There is not a single characteristic that resembles a hammerhead. It is very clearly not the class.

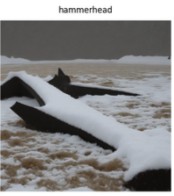

Figure 32: **Set of instructions for labeling.** Instructions provided to the human annotators to perform the labeling of the out-of-class filtering dataset.

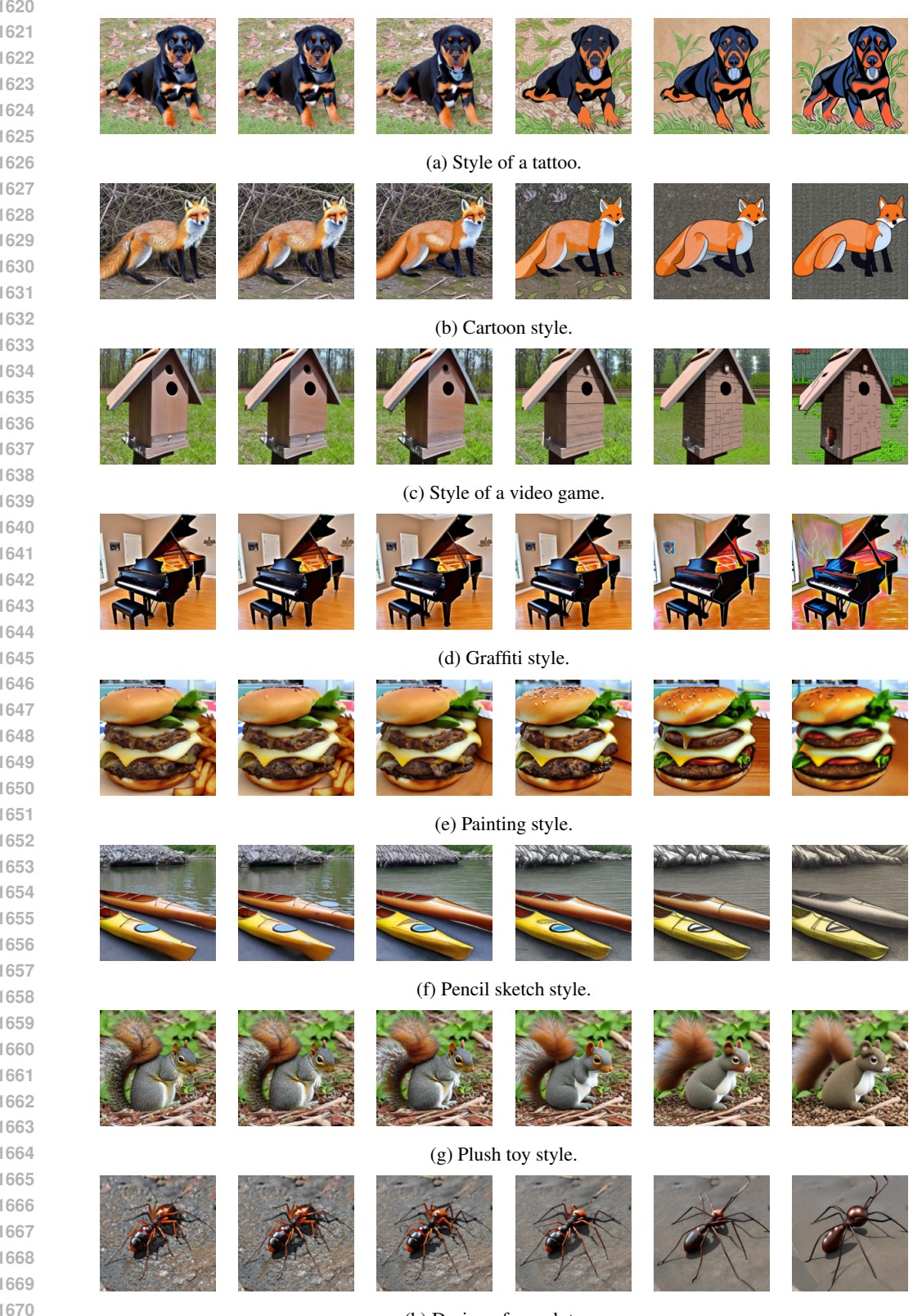

(a) Style of a tattoo.

(b) Cartoon style.

(c) Style of a video game.

(d) Graffiti style.

(e) Painting style.

(f) Pencil sketch style.

(g) Plush toy style.

(h) Design of a sculpture.

Figure 33: **Example sliding for various nuisance shifts.** We visualize six generated images with the corresponding scales as 0, 0.5, 1, 1.5, 2, and 2.5.

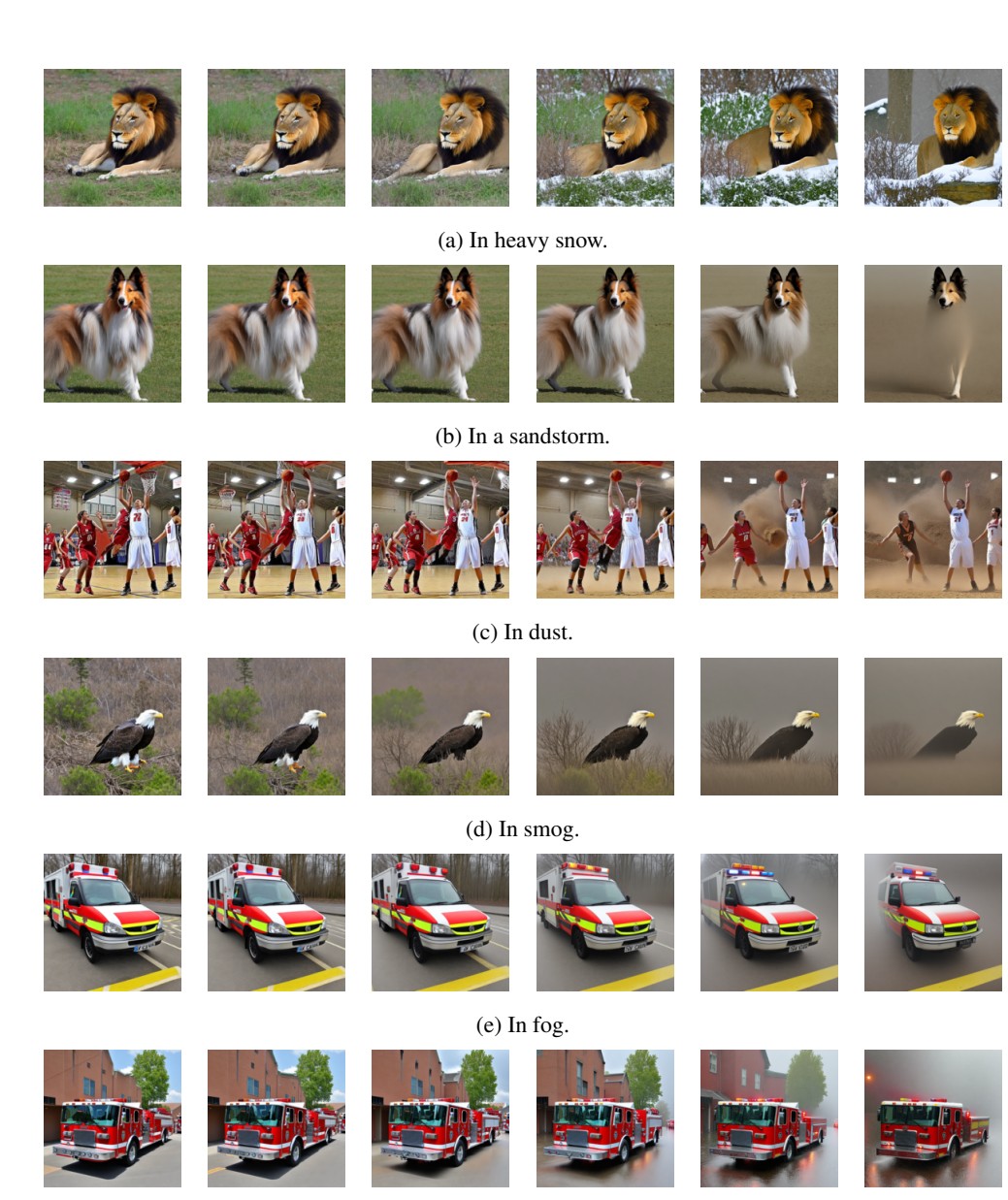

(a) In heavy snow.

(b) In a sandstorm.

(c) In dust.

(d) In smog.

(e) In fog.

(f) In heavy rain.

Figure 34: **Example sliding for various nuisance shifts.** We visualize six generated images with the corresponding scales as 0, 0.5, 1, 1.5, 2, and 2.5.

## B   DATASHEET

In the following, we answer the questions as proposed in Gebru et al. (2021).

### B.1   MOTIVATION

**For what purpose was the dataset created?** Was there a specific task in mind? Was there a specific gap that needed to be filled? Please provide a description.

The dataset was created to evaluate the robustness of state-of-the-art models to specific continuous nuisance shifts. Current approaches are not scalable and often include only a small variety of nuisance shifts, which are not always relevant in the real world. More importantly, current benchmark datasets define binary nuisance shifts by considering the existence or absence of that shift, which may contradict their continuous realization in real-world scenarios.

**Who created the dataset (e.g., which team, research group) and on behalf of which entity (e.g., company, institution, organization)?**

Until the acceptance of the paper, the specific details about the research group, their affiliations, and the entities they represent will remain anonymous.

**Who funded the creation of the dataset?** If there is an associated grant, please provide the name of the grantor and the grant name and number.

Until the acceptance of the paper, the specific details about funding will remain anonymous.

### B.2   COMPOSITION

**What do the instances that comprise the dataset represent (e.g., documents, photos, people, countries)?**

The dataset consists of synthetic images that were generated using Stable Diffusion.

**How many instances are there in total (of each type, if appropriate)?**

The dataset contains $192,168$ images in total, with $32,028$ for each of the six scales with 14 shifts. Each shift has at least $5,000$ images and 100 classes.

**Does the dataset contain all possible instances or is it a sample (not necessarily random) of instances from a larger set?** If the dataset is a sample, then what is the larger set? Is the sample representative of the larger set (e.g., geographic coverage)? If so, please describe how this representativeness was validated/verified. If it is not representative of the larger set, please describe why not (e.g., to cover a more diverse range of instances because instances were withheld or unavailable).

The dataset contains the subset of images that were filtered using the selected filtering strategy. Originally, $420,000$ images were generated.

**What data does each instance consist of? "Raw" data (e.g., unprocessed text or images) or features?** In either case, please provide a description.

"Raw" synthetically generated data as described in the paper.

**Is there a label or target associated with each instance?** If so, please provide a description.

Yes, each image belongs to an ImageNet class and has a shift scale assigned to it.

**Is any information missing from individual instances?** If so, please provide a description, explaining why this information is missing (e.g., because it was unavailable). This does not include intentionally removed information, but might include, e.g., redacted text.

No, for each instance, we give the class label, the scale of the shift, and the parameters used for generating this image. However, the class label might be erroneous in rare cases where the generated image corresponds to an out-of-class sample.

**Are relationships between individual instances made explicit (e.g., users with their tweets, songs with their lyrics, nodes with edges)?** If so, please describe how these relationships are made explicit.

Yes, the relationships in terms of class, random seed for generation, shift, and scale of shift are provided in the dataset.

**Are there recommended data splits (e.g., training, development/validation, testing)?** If so, please provide a description of these splits, explaining the rationale behind them.

We offer a benchmark dataset specifically intended for testing the robustness of classifiers. Therefore, we recommend utilizing the entire dataset provided as the test dataset.

**Are there any errors, sources of noise, or redundancies in the dataset?** If so, please provide a description.

We provided a dataset of generated images. While we apply a filtering strategy to reduce the number of out-of-class and unrealistic samples, we cannot guarantee that all images of the dataset represent a realistic and visually appealing realization of the considered class. We provide a statistical estimate of the number of failure samples in the paper. The data might also include the redundancies that underlie the image generation process of Stable Diffusion.

**Is the dataset self-contained, or does it link to or otherwise rely on external resources (e.g., websites, tweets, other datasets)?** If it links to or relies on external resources, a) are there guarantees that they will exist, and remain constant, over time; b) are there official archival versions of the complete dataset (i.e., including the external resources as they existed at the time the dataset was created); c) are there any restrictions (e.g., licenses, fees) associated with the use of these external resources?

The dataset is fully self-contained.

**Does the dataset contain data that might be considered confidential (e.g., data that is protected by legal privilege or by doctor–patient confidentiality, data that includes the content of individuals' non-public communications)?** If so, please provide a description.

No.

**Does the dataset contain data that, if viewed directly, might be offensive, insulting, threatening, or might otherwise cause anxiety?** If so, please describe why.

There is a small chance that our synthetically generated data can generate offensive images. However, we did not encounter any such sample during our extensive manual annotations.

**Does the dataset relate to people? If not, you may skip the remaining questions in this section.**

No.

**Does the dataset identify any subpopulations (e.g., by age, gender)?** If so, please describe how these subpopulations are identified and provide a description of their respective distributions within the dataset.

N/A.

**Is it possible to identify individuals (i.e., one or more natural persons), either directly or indirectly (i.e., in combination with other data) from the dataset?** If so, please describe how.

N/A.

**Does the dataset contain data on individuals' protected characteristics (e.g., age, gender, race, religion, sexual orientation)?** If so, please describe this data and how it was obtained.

N/A.

**Does the dataset contain data on individuals' criminal history or other behaviors that would typically be considered sensitive or confidential?** If so, please describe this data and how it was obtained.

N/A.

### B.3 COLLECTION PROCESS

**How was the data associated with each instance acquired? Was the data directly observable (e.g., raw text, movie ratings), reported by subjects (e.g., survey responses), or indirectly inferred/derived from other data (e.g., part-of-speech tags, model-based guesses)?**

N/A.

**What mechanisms or procedures were used to collect the data (e.g., hardware apparatus or sensor, manual human curation, software program, software API)? How were these mechanisms or procedures validated?**

We used Stable Diffusion 2.0 to generate all images. Images were generated using NVIDIA A100 and A40 GPUs.

**If the dataset is a sample from a larger set, what was the sampling strategy (e.g., deterministic, probabilistic with specific sampling probabilities)?**

The dataset was filtered using a combinatorial selection approach using the alignment scores of DINOv2 and CLIP to the considered class.

**Who was involved in the data collection process (e.g., students, crowdworkers, contractors) and how were they compensated (e.g., how much were crowdworkers paid)?**

The authors of the paper and other PhD students of the institute. They were not additionally paid for the dataset collection process.

**Over what timeframe was the data collected? Does this timeframe match the creation timeframe of the data associated with the instances (e.g., recent crawl of old news articles)?** If not, please describe the timeframe in which the data associated with the instances was created.

The images were generated and processed over a timeframe of four weeks.

**Were any ethical review processes conducted (e.g., by an institutional review board)?** If so, please provide a description of these review processes, including the outcomes, as well as a link or other access point to any supporting documentation.

No ethical concerns.

### B.4 PREPROCESSING/CLEANING/LABELING

**Was any preprocessing/cleaning/labeling of the data done (e.g., discretization or bucketing, tokenization, part-of-speech tagging, SIFT feature extraction, removal of instances, processing of missing values)?** If so, please provide a description. If not, you may skip the remaining questions in this section.

Yes, cleaning of the generated data was conducted. The generated images underwent filtering to reduce the number of out-of-class samples using the proposed filtering mechanisms. Instances that did not meet these criteria were removed from the dataset. For a detailed description of the filtering process, please refer to the corresponding section in the paper.

**Was the "raw" data saved in addition to the preprocessed/cleaned/labeled data (e.g., to support unanticipated future uses)?** If so, please provide a link or other access point to the "raw" data.

The generated images remain in their original, unprocessed state and can be considered as "raw" data. However, we have not provided all the images that were filtered out.

**Is the software used to preprocess/clean/label the instances available?** If so, please provide a link or other access point.

Generating the images was performed using commonly available Python libraries. For annotating a subset of the dataset for filtering purposes, we have used the VIA annotation tool (Dutta & Zisserman, 2019; Dutta et al., 2016).

## B.5 USES

**Has the dataset been used for any tasks already?** If so, please provide a description.

In our work, we demonstrate how this approach yields valuable insights into the robustness of state-of-the-art models, particularly in the context of classification tasks.

**Is there a repository that links to any or all papers or systems that use the dataset?** If so, please provide a link or other access point.

We will provide a link that includes all relevant papers or systems.

**What (other) tasks could the dataset be used for?**

Our work showcases the capability of our dataset to enhance control over data generation, which is particularly evident through continuous shifts. However, its applicability extends beyond this demonstration. The dataset can be effectively utilized in various generation tasks that necessitate continuous parameter control. While we showcased its efficacy in providing insights for models tackling classification tasks, it can seamlessly extend to evaluate the robustness of state-of-the-art methods across diverse tasks such as segmentation, domain adaptation, and many others. This is possible by combining our approach with other modes of conditioning Stable Diffusion. In addition, our data can also be used for fine-tuning, which we also demonstrated in the supplementary material.

**Is there anything about the composition of the dataset or the way it was collected and cleaned that might impact future uses? For example, is there anything that might cause the dataset to be used inappropriately or misinterpreted (e.g., accidentally incorporating biases, reinforcing stereotypes)?**

Our dataset was synthesized using a generative model. It, therefore, likely inherits any biases for its generator. Similarly, filtering is performed by pre-trained models, which can indirectly also contribute to biases.

**Are there tasks for which the dataset should not be used?** If so, please provide a description.

No, there are no tasks for which the dataset should not be used. Our dataset aims to enhance model robustness and provide deeper insights during model evaluation. Therefore, we see no reason to restrict its usage.

## B.6 DISTRIBUTION

**Will the dataset be distributed to third parties outside of the entity (e.g., company, institution, organization) on behalf of which the dataset was created?** If so, please provide a description.

Yes, the dataset will be publicly available on the internet.

**How will the dataset be distributed (e.g., tarball on website, API, GitHub)? Does the dataset have a digital object identifier (DOI)?**

In the future, we will distribute the dataset as a tarball on our servers.

**When will the dataset be distributed?**

The dataset will be distributed upon acceptance of the manuscript. It is now available under the provided anonymized link.

**Will the dataset be distributed under a copyright or other intellectual property (IP) license, and/or under applicable terms of use (ToU)?** If so, please describe this license and/or ToU, and provide a link or other access point to, or otherwise reproduce, any relevant licensing terms or ToU.

CC-BY-4.0.

**Have any third parties imposed IP-based or other restrictions on the data associated with the instances?** If so, please describe these restrictions, and provide a link or other access point to, or otherwise reproduce, any relevant licensing terms.

No, there are no IP-based or other restrictions on the data associated with the instances imposed by third parties.

**Do any export controls or other regulatory restrictions apply to the dataset or to individual instances?** If so, please describe these restrictions, and provide a link or other access point to, or otherwise reproduce, any supporting documentation.

We are not aware of any export controls or other regulatory restrictions that apply to the dataset or to individual instances.

## B.7 MAINTENANCE

**Who is supporting/hosting/maintaining the dataset?**

The dataset is supported by the authors and their associated research groups. The dataset is hosted on our own servers.

**How can the owner/curator/manager of the dataset be contacted (e.g., email address)?**

The authors of this dataset will be reachable at their e-mail addresses: [undisclosed]. In addition, we will add a contact form, which will be made available on the website.

**Is there an erratum?** If so, please provide a link or other access point.

If errors are found, an erratum will be added to the website.

**Will the dataset be updated (e.g., to correct labeling errors, add new instances, delete instances)?** If so, please describe how often, when, and how updates will be provided.

Yes, updates will be communicated via the website. The dataset will be versioned.

**If the dataset relates to people, are there applicable limits on the retention of the data associated with the instances (e.g., were individuals in question told that their data would be retained for a specific period of time and then deleted)?** If so, please describe these limits and explain how they will be enforced.

Our dataset does not relate to people.

**Will older versions of the dataset continue to be supported/hosted/maintained?** If so, please describe how.

No, older versions of the dataset will not be supported if the dataset is updated. We do not plan to extend or update the dataset. Any updates will be made solely to correct any hypothetical errors that may be discovered.

**If others want to extend/augment/build on/contribute to the dataset, is there a mechanism for them to do so?** If so, please provide a description. Will these contributions be made publicly available?

Yes, we provide all the necessary tools and explanations to enable users to build continuous shifts for their own specific applications. Our dataset serves as a foundation to illustrate how it can be used to evaluate current state-of-the-art methods. However, we are happy to centralize and showcase all related work on our GitHub page that benefits from our method of generating data.

## B.8 AUTHOR STATEMENT OF RESPONSIBILITY

The authors confirm all responsibility in case of violation of rights and confirm the license associated with the dataset and its images.

