# OpenReview forum: "CNS-Bench: Benchmarking Model Robustness Under Continuous Nuisance Shifts"
_ICLR.cc/2025/Conference — Submitted to ICLR 2025_

### Official Review · Reviewer_1iK7 · 2024-10-27

**Soundness:** 3
**Presentation:** 3
**Contribution:** 2
**Rating:** 6
**Confidence:** 4

**Summary:**

This paper introduces a novel benchmark for evaluating the OOD robustness of vision models. The core idea is to build a system that can generate images from the training distribution, but with natural distribution shifts (like snow) applied *with continuous severity levels*, so that one can smoothly increase the degree of corruption. The authors achieve this by leveraging diffusion models conditioned on the training distribution in combination with LoRA adapters. The resulting benchmark does therefore not only yield scalar accuracy values, but performance curves for different models, relating the severity of the corruption to the drop in classification performance.

**Strengths:**

The paper seems technically sound and successfully combines different existing methods to achieve the stated goal of generating a benchmark of continuous distribution shifts. I appreciate the thorough analysis and sanity-checks, such as creating a large OOC-detection dataset to make sure that the proposed filtering mechanism works. The writing is mostly clear, although some questions remain (see below). As far as I can tell (although I'm not too familiar with generative models) the authors cite the relevant related work.

**Weaknesses:**

One fundamental weakness of the paper is the lack of motivation for why a robustness evaluation at different levels is important. I’m aware that ImageNet-C also offers different corruption levels, and I could maybe be convinced that having access to these levels is useful, but the analyses conducted here do not really achieve this: Why is it interesting at which severity level a model fails, especially given that it’s unclear whether the corruption severity levels across different shifts and different classes are properly calibrated against each other (see my question 4)? Of course, having a method of subjecting any training set to a natural distribution shift is great, but the Dataset Interface paper already achieves this. So the overall contribution of the paper is effectively “only” interpolating between uncorrupted images and fully corrupted images, but I wonder why that matters, unless the ordering of models drastically changes across the different levels. That does not seem to be the case overall, according to figure 6a, and I wonder whether the differences in figure 6b and 6c (where values are averaged over fewer trials) are statistically stable. Adding confidence intervals to these plots would help convince me that this is indeed a robust finding. But even if this were the case: If I had a dataset with a painting-corruption, how would I know what the corruption-scale of my dataset is, to then select the best model at that level? And do I really care about the minuscule differences between models (<< 1% accuracy delta) at scale 1, or would I simply select the model that does best at the maximum scale?
While I appreciate that the authors included the failure cases in figure 16, they do make me wonder how reliably the method really works, and whether this unreliability might explain the weird curves in figure 6c. It would be good to also add confidence intervals to figure 3, to give a better idea of the quality of the generated images (but see my question 2 about the y-axis values of figure 3).

**Questions:**

## Feedback
* I would have liked to take a closer look at the images in the benchmark, but could not unzip the provided benchmark.zip file, apparently because the file was corrupted. I don't think it's an issue on my end, could you look into this?
* I think the writing, especially in section 3.2 where the method is explained, could be improved quite a bit, also to render the paper more self-sustained - I found myself having to look up the referenced papers, even though the relevant parts could have been summarized in a few sentences. For example, how exactly the scale of the sliders works cannot be understood from this paper alone, one needs to read Gandikota et al. 2023.
* The legend of figure 7 is broken. The label for scale 1.5 appears twice and the values are not ordered.
* Minor point, but in figures 9 and 10 it might be better to share the y-axis for more comparability between the plots.

## Questions
1. In line 197, shouldn’t $\theta$ have both $c_t$ and $c_+$ in the subscript, like $\theta_{c_t, c_+}$?
2. In figure 3, how is it possible that the difference of two cosine similarities, which should be <= 2, achieves values of up to 7.5?
3. In line 423, you write that an explanation for the abrupt change in failure rate of the cartoon style is the ImageNet class “comic book”, but I don’t see why images would be mis-classified as comic books more for scale 1.5 than for scale 2 and higher.
4. Do you have any way of asserting that the severity levels of different shifts and different classes are actually calibrated, i.e. that scale 2.5 of an elephant in snow is the same level of corruption as a scale 2.5 zebra in fog? Since you are training different LoRAs for the different classes, I’m not sure if this will always be the case, but it might be desirable. (I guess one could calibrate this using the CLIP-distances…?)
5. In principle, could you combine different distribution shifts at the same time? E.g., modify the same image to both exhibit fog and snow?

## Final Assessment
Overall, I’m a bit skeptical of the relevance of the contribution of the paper (see above) and could not check how the images in the benchmark look like, qualitatively. I propose to reject for now, but I'm curious to hear the perspectives of the other reviewers and would be willing to increase my score if they deem this work relevant, or if the authors can motivate the need for continuous shifts better.

---

> ### Author Response · Authors · 2024-11-21
> **Response to reviewer 1iK7**
>
> > One fundamental weakness of the paper is the lack of motivation for why a robustness evaluation at different levels is important.
>
> We refer to the common response (i) where we discuss that considering multiple scales allows a more nuanced study of model robustness.
>
> > Why is it interesting at which severity level a model fails, especially given that it’s unclear whether the corruption severity levels across different shifts and different classes are properly calibrated against each other?
>
> Calibrating shifts is difficult or subjective and maybe even impossible across different shifts, i.e., sand storm and painting, which are fundamentally different. Similarly, not all levels of different corruptions types in ImageNet-C can be directly compared. Therefore, our motivation for the failure point is grounded by the goal to check whether a model fails earlier or later than other models, which does not require a calibration across shifts but still enables us to already make important observations.
>
> > Of course, having a method of subjecting any training set to a natural distribution shift is great, but the Dataset Interface paper already achieves this.
>
> We agree that the Dataset Interface provides a valuable basis for generative benchmarking. However, we believe that the contribution of being able to realizing continuous shifts is note-worthy since it allows a more nuanced and systematic study of robustness, showing that the ordering of models can change for different scales and shifts. Additionally, we enable more fine-granular changes by applying LoRA adapters. And lastly, we carefully analyze the effect of out-of-class samples and propose a more effective mechanism that better removes such OOC samples.
>
> > but I wonder why that matters, unless the ordering of models drastically changes across the different levels.
> > I wonder whether the differences in figure 6b and 6c [...] are statistically stable.
>
> To address your concern whether our findings are statistically stable, we plot the confidence intervals in Fig. 13 and 14, which shows that some of the depicted model rankings actually change significantly.
> Therefore, we underline that aggregating the performance into one single metric removes relevant insights when evaluating the OOD robustness in a specific scenario.
>
> > If I had a dataset with a painting-corruption, how would I know what the corruption-scale of my dataset is, to then select the best model at that level?
>
> We agree that our work does not address this question. While we focus on benchmarking along our pre-defined scales, estimating the shift scale of a specific image or dataset is a fundamentally different task that goes beyond the scope of our work. However, our trained LoRA adapter might be potentially applicable for this task, which is an exciting future direction. The adapters scales could be estimated using the standard diffusion noise prediction objective, providing a measure of the average shift scale of a dataset or even a single image.
>
> > And do I really care about the minuscule differences between models (<< 1% accuracy delta) at scale 1, or would I simply select the model that does best at the maximum scale?
>
> Ideally, one would choose a model that performs best for all scales. However, we, again, follow the same motivation as ImageNet-C (see common response (i)): If one chooses simply the model with the best performance for the largest scale, one might end up using a model that performs worse for all other scales.
> Let's consider a specific example for the sandstorm shift (as in Fig. 34b): A model that only focuses on the front perspective of the head of the dog, will outperform a model at the last scale even though it might perform worse for most other images of that class.
>
> > While I appreciate that the authors included the failure cases in figure 16, they do make me wonder how reliably the method really works [...] It would be good to also add confidence intervals to figure 3.
>
> Fig. 16 (Fig. 26 in the updated manuscript) shows failure cases of the naive strategy for achieving continuous shifts (interpolation of text embeddings) and explains why we did not pursue this simple strategy and, instead, worked with LoRA adapters. An example failure case of our method is depicted in Fig. 2, which our filter strategy removes. As stated in the common response (iii), we conducted a human assessment that showed that 1% of our images are not samples of the class. We hope this also addresses your concerns. We plot the confidence intervals for Fig. 3 in Fig. 15.
>
> > I would have liked to take a closer look at the images in the benchmark, but could not unzip the provided benchmark.zip file, apparently because the file was corrupted. I don't think it's an issue on my end, could you look into this?
>
> The `benchmark.zip` file works on our end, but Google may show warnings with large files. We’ve uploaded a subset of the dataset in `CNS_dataset/example_imgs` on Google Drive. We hope this helps!

---

> ### Author Response · Authors · 2024-11-21
> **Response 2 to reviewer 1iK7**
>
> > I think the writing, especially in section 3.2 where the method is explained, could be improved quite a bit.
>
> We rewrote that section and are open to any additional feedback about that part.
>
> > The legend of figure 7 is broken. [...] in figures 9 and 10 it might be better to share the y-axis for more comparability between the plots.
>
> Thanks for pointing out that. We addressed the point in the updated manuscript.
>
> > Subscripts in line 197
>
> We reworked the whole paragraph and we hope this addresses your comment.
>
> > In figure 3, how is it possible that the difference of two cosine similarities, which should be <= 2, achieves values of up to 7.5?
>
> Thank you for raising this inconsistency. For this plot, we multiplied the cosine similarities by 100. We updated the plot accordingly.
>
> > In line 423, you write that an explanation for the abrupt change in failure rate of the cartoon style is the ImageNet class “comic book”, but I don’t see why images would be misclassified as comic books more for scale 1.5 than for scale 2 and higher.
>
> First, we quantitatively evaluated our empirical observations and we report the ratio of classes that were wrongly classified as comic book for the cartoon shift:
>
> | Scale | Ratio |
> | -------- | -------- |
> | 0     | 0.00     |
> | 0.5     | 0.00     |
> | 1.     | 0.02     |
> | 1.5     | 0.13     |
> | 2.     | 0.23     |
> | 2.5     | 0.32     |
>
> These evaluations show that a significant part of wrong classifications (>50%) can be, indeed, attributed to the comic-book class.
> This also shows that the misclassifications increase: While the point of style change seems to be rather abrupt for a human, some visual properties continue to shift towards a more simplistic cartoon at later steps. Therefore, more and more images are misclassified.
>
>
> > Do you have any way of asserting that the severity levels of different shifts and different classes are actually calibrated? Since you are training different LoRAs for the different classes, I’m not sure if this will always be the case, but it might be desirable. (I guess one could calibrate this using the CLIP-distances…?)
>
> We refer to our answer to your previously raised point about the calibration to address this comment (2nd point in this comment). Calibrating shifts is an exciting challenge for future work. Applying CLIP might be good way to achieve this. However, the CLIP measure is not always reliable for our structure-preserving shifts, as we exemplarily visualize in Fig. 16.
>
> > In principle, could you combine different distribution shifts at the same time? E.g., modify the same image to both exhibit fog and snow?
>
> Yes, our LoRA adapters can be composed, as demonstrated in an example application in the supplementary material (Fig. 29). Exploring the robustness to combined nuisance shifts is indeed an interesting direction for future work.
>
> Additionally, we would like to highlight a broader aspect of our method: it is not limited to nuisance shifts specified via text. Our approach has the potential to handle other types of nuisance shifts, such as a continuous distribution shift from ImageNet to ImageNetv2. However, for this work, we choose to focus specifically on more confined distribution shifts to maintain a clear scope.

---

> > ### Comment · Reviewer_1iK7 · 2024-11-26
> > **What exactly do you mean by accuracy drop?**
> >
> > Dear Authors, \
> > thank you for your detailed response and for providing the more easily accessible example images. I find them very helpful to better understand the quality of the generated distribution shifts. For some shifts, I find the quality of the images convincing (e.g. cartoon style). **However, the “dust”, “rain”, “sandstorm”, “fog”, and “snow” categories do in my opinion not pose an OOD scenario at all, since they do not even occlude the object, but merely change the background slightly.** This came as a surprise to me, because I don't see how these images would be harder for any reasonable model than uncorrupted images. This made me realize that the *accuracy drop* you refer to in e.g. figure 5 is the **absolute delta in classification performance**, not the relative one. In other words, the accuracy of a ResNet-50, which should sit at a performance of about 80%, drops by 0.6 *percentage points* (e.g. to an accuracy of 79.4%) on the largest scale of your dataset, is that correct? Are you seriously proposing an "OOD-benchmark" for which even the hardest corruptions reduce the performance of a vanilla ResNet-50 by less than 1%? Please clarify this point, as otherwise I see myself **forced to change my rating to a high-confidence clear reject, since the method simply does not work**.
> >
> > “Calibrating shifts is difficult or subjective and maybe even impossible across different shifts” \
> > I agree, hence my question about this - I don’t think it is legitimate to calculate the failure point by averaging over the different shifts, since the different shifts could be on very different failure scales.
> >
> > “we plot the confidence intervals in Fig. 13 and 14” \
> > Thank you! I appreciate this, and find the confidence intervals smaller than expected (I assume this is a 95% confidence interval?)
> >
> > “Fig. 16 (Fig. 26 in the updated manuscript) shows failure cases of the naive strategy for achieving continuous shifts” \
> > My apologies, I had missed that. Thank you for also plotting the confidence intervals in figure 3.
> >
> > “We rewrote that section [3.1] and are open to any additional feedback about that part.” \
> > Currently, section 3.1 still is not self-containing, as somebody unfamiliar with the work of Vendrow et al 2023 and Gal et al 2023 will have a hard time understanding what was done. It would not hurt to explain a bit more of what was done here. In line 194 there’s a “that” which doesn’t belong there.
> >
> > “a significant part of wrong classifications (>50%) can be, indeed, attributed to the comic-book class” \
> > That’s interesting, thanks. What I was wondering about was why the FP ratio would decrease for the larger scales again (I would have expected the results to look like your table does, with more mis-classifications at the larger scales).
> >
> > “Yes, our LoRA adapters can be composed, as demonstrated in an example application in the supplementary material (Fig. 29)” \
> > Cool, thank you!
> >
> > “Additionally, we would like to highlight a broader aspect of our method: it is not limited to nuisance shifts specified via text.” \
> > I agree that this is a compelling feature of your method.

---

> ### Author Response · Authors · 2024-11-27
> **Clarification of accuracy drop**
>
> Thank you again for your response. We will address your comments individually:
>
> > [...] the accuracy drop you refer to in e.g. figure 5 is the absolute delta in classification performance, not the relative one. In other words, the accuracy of a ResNet-50, which should sit at a performance of about 80%, drops by 0.6 percentage points (e.g. to an accuracy of 79.4%) on the largest scale of your dataset, is that correct? Are you seriously proposing an "OOD-benchmark" for which even the hardest corruptions reduce the performance of a vanilla ResNet-50 by less than 1%? Please clarify this point.
>
> We are sorry for the misunderstanding and we would like to clarify the scale of the y-axis: The y-axis is, indeed, the absolute performance drop. However, we do not report in % but in the value range [0,1] throughout our whole analysis. Therefore, the correct interpretation of the graph is that, for example, ResNet drops by 60 percentage points for the nuisance shift 'cartoon' (i.e. 0.6).
> We updated the caption of the figure accordingly and also mentioned it explicitly in the experimental details (*Metrics*) to ensure a proper understanding.
>
> > The “dust”, “rain”, “sandstorm”, “fog”, and “snow” categories do in my opinion not pose an OOD scenario at all, since they do not even occlude the object, but merely change the background slightly.
>
> Occlusions of objects are indeed an interesting OOD factor to consider. However, OOD scenarios can also include many other factors of distribution shifts, e.g., texture variations, high frequency corruptions or background changes.
> While the applied nuisance shifts in terms of weather conditions are sometimes not as drastic as one might expect, please note that their application already significantly reduces the performance of the classifiers (e.g., 20% for ResNet-50 for snow) and, hence, clearly induces an out-of-distribution shift of the data. Please also consider that the presented subset includes images with varying scales and some images actually include (partial) occlusions.
>
> > This came as a surprise to me, because I don't see how these images would be harder for any reasonable model than uncorrupted images.
>
> It is a known phenomenon that classifiers make mistakes that are rather unexpected for humans. For example, it was shown that classifiers can fail to recognize objects due to simple changes that would not fool a human observers, such as changes in the object pose [A], the background [B], or the texture of the object [C]. This observation is commonly attributed to the "shortcut learning" phenomenon [D], where models rely on simple, often misleading patterns in the data, rather than understanding the underlying concepts.
> In this context, our framework is the first that enables the systematic evaluation with respect to diverse and continuous real-world nuisance shifts, providing deeper insights into model robustness.
>
> [A] Michael A. Alcorn, Qi Li, Zhitao Gong, Chengfei Wang, Long Mai, Wei-Shinn Ku, and Anh Nguyen. "Strike (with) a pose: Neural networks are easily fooled by strange poses of familiar objects." Proceedings of the IEEE/CVF conference on computer vision and pattern recognition. 2019.
>
> [B] Rosenfeld, Amir, Richard Zemel, and John K. Tsotsos. "The elephant in the room." arXiv preprint arXiv:1808.03305. 2018.
>
> [C] Robert Geirhos, Patricia Rubisch, Claudio Michaelis, Matthias Bethge, Felix A. Wichmann, and Wieland Brendel. "ImageNet-trained CNNs are biased towards texture; increasing shape bias improves accuracy and robustness." International Conference on Learning Representations. 2018.
>
> [D] Robert Geirhos, Jörn-Henrik Jacobsen, Claudio Michaelis, Richard Zemel, Wieland Brendel, Matthias Bethge, and Felix A. Wichmann. "Shortcut learning in deep neural networks." Nature Machine Intelligence 2.11: 665-673. 2020.

---

> ### Author Response · Authors · 2024-11-27
> **Answers to further remarks**
>
> > Calibrating shifts is difficult [...] I agree, hence my question about this - I don’t think it is legitimate to calculate the failure point by averaging over the different shifts, since the different shifts could be on very different failure scales.
>
> We agree with the concern of the reviewer and removed results with averaged failure points (Tab. 4 in the previous manuscript). Please note that we motivated the failure point computation for individual shifts (l. 465-472).
>
>
> > I appreciate this, and find the confidence intervals smaller than expected (I assume this is a 95% confidence interval?)
>
> The plots in Fig. 13 and 14 depict the one-sigma confidence interval, as mentioned in the caption. To address your original comment whether model rankings actually change significantly, we perform a statistical test: We test whether the estimated accuracy drops are significantly different using the two proportion z-test and a p-level of 0.05. For example, for the painting style nuisance we find that the accuracy drop differences between RN152 and ViT, ConvNext and ViT, DeiT and ViT, DeiT3 and ViT significantly change the sign when increasing the severity of the shift.
> We further checked which shifts result in significant model ranking changes for the considered models along the architecture axis. We observe such statistically significant changes for the following shifts: painting style, the style of a tattoo, heavy sandstorm.
>
>
> > Currently, section 3.1 still is not self-containing, as somebody unfamiliar with the work of Vendrow et al 2023 and Gal et al 2023 will have a hard time understanding what was done. It would not hurt to explain a bit more of what was done here. In line 194 there’s a “that” which doesn’t belong there.
>
> Following your initial feedback, we specifically addressed the criticism about the missing clarity of Sec. 3.2. Now, we also updated section Sec. 3.1. We hope this revised version is clearer and we are looking for your feedback. We are willing to add a more formal explanation of the diffusion model objective if you think, this should not be missing.
>
> Please let us know if you see potential for more clarity in other parts of the paper. We are happy to address these points as well. Thank you for your thorough and constructive feedback!
>
> > That’s interesting, thanks. What I was wondering about was why the FP ratio would decrease for the larger scales again (I would have expected the results to look like your table does, with more mis-classifications at the larger scales).
>
> The failure point distribution quantifies when a classifier *starts* failing (per scale), which rather relates to the performance drop with respect to the previous scale. In contrast, the presented table measures the absolute accuracy drop per scale with respect to scale 0. In that case, the drop from scale 1 to scale 1.5 is the largest (0.13-0.02=0.11). That means, the classifier starts failing more often at intermediate scales for such shifts.

---

> > ### Comment · Reviewer_1iK7 · 2024-12-01
> >
> > Dear Authors, \
> >
> > thank you for the clarifying response about the y-axis scale for the accuracy drop, I am relieved.
> >
> > “please note that their application already significantly reduces the performance of the classifiers (e.g., 20% for ResNet-50 for snow)” \
> > According to figure 9, the accuracy of the (ImageNet-trained) RN-152 hardly changes up until scale 1.5 for corruptions snow, fog, smog and rain. Even under the correct interpretation of the y-axis, this seems like a rather small reduction in accuracy. But fair point, the fact that the accuracy changes at all for the larger scales potentially reveals that models rely on spurious background features.
> >
> > “We agree with the concern of the reviewer and removed results with averaged failure points.” \
> > I appreciate this - calculating FPs only for each shift seems more reasonable.
> >
> > “To address your original comment whether model rankings actually change significantly, we perform a statistical test” \
> > Thank you for conducting this analysis, good idea!
> >
> > “Now, we also updated section Sec. 3.1” \
> > Thanks, looks better!
> >
> > To summarize my final opinion on the paper, I think that the suggested methodology, while not entirely novel, could prove to be useful, especially to simulate certain new distribution shifts that would be otherwise hard to simulate. This warrants an increase of my score to a 6, even though the benchmark itself still fails to convince me, because its marginal utility seems fairly low -- mainly because the images just seem too easy and the drops in accuracy (even if correctly interpreted; my apologies for misunderstanding this earlier) are rather modest, even for ImageNet-trained models. I'm afraid that in the era of models trained on web-scale datasets, the utility of this specific benchmark might be rather low and I doubt that it will see much use, but the methodology is valuable work that others could build on.

---

> ### Author Response · Authors · 2024-12-04
> **Thank you for your review**
>
> Dear reviewer 1iK7,
>
> thank you for your response and your active engagement in our discussion. This clearly helped us improve our paper.
> We are grateful that you recognize the significance of our work and you agree with the other reviewers that our work represents a valuable contribution to the community.
>
> The Authors

---

### Official Review · Reviewer_zdn7 · 2024-10-29

**Soundness:** 3
**Presentation:** 3
**Contribution:** 3
**Rating:** 6
**Confidence:** 4

**Summary:**

This paper introduces CNS-Bench, a benchmark for evaluating the robustness of image classifiers to what the authors call "continuous nuisance shifts" - essentially OOD distortions like no snow -> snow along a continuous axis. CNS-Bench uses LoRA adapters applied to diffusion models to generate images with a wide range of nuisance shifts at various severities. While in principle continuous shifts are possible, most of the article nevertheless focuses on a fixed number of shifts (5 severity levels). The authors then conducted an evaluation of few different visual image classifier families on CNS-Bench.

The paper's contributions are defined, by the authors, as follows:
1. The creation of CNS-Bench & evaluation of models
2. The collection of an annotated dataset for filtering (note: this is a process that becomes necessary since the approach used in the paper may alter the class label, therefore this essentially fixes an issue introduced by the approach)
3. The publication of 14 nuisance shifts at five severity levels. (note: this is essentially part of #1)

**Strengths:**

- Authors promise to release the dataset under a permissive licences (CC-BY-4.0); code is available from supplementary material via Google Drive.
- I like the approach of measuring a precise failure point. In psychophysics, a related concept is called the threshold of a model - see, e.g., Figure 4 of this 2017 paper on "object recognition when the signal gets weaker": https://arxiv.org/pdf/1706.06969. A threshold is calculated across many samples; the failure point described in this article, in contrast, is the point where an individual test sample is no longer correctly recognized.
- The technical approach is a nice, simple and creative application of generative diffusion models.

**Weaknesses:**

1. Nuisance shifts affect information that's not related to the nuisance concept. In Figure 22 and 23, some nuisance shifts don't achieve the desired result; e.g. the variation "in rain" (Fig 23f) alters/blurs the background without occluding the object through rain. **Some nuisance shifts introduce confounds**, e.g. "in dust" not only adds dust but also removes half of the people in the image and changes a person's shirt color from red to black. As a consequence, failures cannot be attributed to the nuisance concept itself.


2. The approach is based on generative models, thereby introducing a **real vs. synthetic distribution shift** that may further influence results. A discussion - better yet: an analysis - of this likely confound is recommended. Without this, I'm hesitant to share the author's hope that ("this benchmark can encourage the community to adopt generated images for evaluating the robustness of vision models.").


3. **The paper's main claim to fame remains a bit unclear to me**, and that's my most important concern. At the same time, this might be the biggest opportunity for improvement and clarification from which future readers might benefit. The authors propose a variety of options to choose from, but I'm not convinced (yet - happy to be convinced of the opposite). Specifically:
- Is it about continuous shifts? If so, this can be achieved with parametric distortions too (e.g. Gaussian noise with noise strength as a continuous parameter). Furthermore, the authors end up narrowing it down to 5 severity levels anyways, which is roughly in line with the 5-8 levels from related work.
- Is it about a large number of distortions? Probably not, since the dataset's 14 distortions are in the same ballpark as ImageNet-C (15 test + 4 validation distortions) or model-vs-human (17 distortions).
- Is it about testing a variety of models? While a number of model families are investigated (CLIP, ConvNext, Deit, Dino, MAE, MOCO, ResNet, ViT) that's also similar to previous investigations, some of which tested a broader variety.
- Is it about identifying failure cases? If so, when is it important to know about a specific failure case (as opposed to a model's threshold, averaged across many samples)?
- Is it about the connection between architecture and robustness? The observation that architecture influences model robustness has been reported (extensively) by a range of previous work.
- Is it about precise control? While strength can be controlled, the effect introduced by the nuisance can't be controlled to a level where no confounds would be introduced, as seen in Figures 22 & 23.
- Is it about scalability? If so, why is training separate LoRA adapters for each ImageNet class and shift more scalable than existing approaches?
- Is it about real-world nuisance shifts? If so, see concern #2 on the real vs. synthetic distribution shift.

I recommend that the authors clearly state and justify what they believe is the primary novel contribution ("claim to fame") of their work, and how it advances the field beyond existing benchmarks and approaches.

**Questions:**

- If it should be a benchmark, people will want to know: who won? What's the score that I need to beat in order to be SOTA? Tables with an overall score would help. Table 1 is a step in the right direction but it's not immediately clear which model is best and which score (Accuracy? Accuracy drop?) is the benchmark score.
- Why were those 14 "nuisances" chosen and not others, why 14 and not, say, 50? (Not saying that the authors should do this but asking out of curiosity)
- What's the robustness of a failure point to random variation?
- Is performance (accuracy) always a monotonous function of the LoRA slider strength? Are there instances when that's not the case? If so, what does it mean if there are images beyond the failure point that are again correctly recognized?
- line 43: "such approaches are not scalable" - why not? If one takes a large dataset and applies cheap corruptions like the ones from ImageNet-C, should that be considered less scaleable?
- What's the computational cost of generating the dataset?

MISC:
- Figure 7: instead of re-using colors that were used in e.g. Figure 6 with a different association, I'd recommend using different colors here to avoid confusion - ideally a sequential color palette, with the legend sorted by scale not arbitrarily. Also, label 1.5 appears twice which is probably not intentional.

---

> ### Author Response · Authors · 2024-11-21
> **Response to reviewer zdn7**
>
> > Nuisance shifts affect information that's not related to the nuisance concept. [...] As a consequence, failures cannot be attributed to the nuisance concept itself.
>
> Our strategy allows applying shifts that were not possible previously. However, using a generative model for realizing shifts inherently comes along with confounders. This is explained by the biases in the training data of the generative model. So, similarly, real-world OOD datasets also exhibit such confounders and it is equally hard to differentiate whether an accuracy drop can be, e.g., explained by
> - a style change or a change of shape as in the ImageNet-R dataset (consider Fig. 24 for examples) or
> - the heavy snow or the lower visual quality in the OOD-CV dataset (as already depicted in Fig. 31).
>
> In contrast to previous works, we particularly reduce the confounders of the classified object by applying LoRA adapters only to later noise steps throughout the diffusion process, which prevents significant changes of spatial structure of the image. Therefore, while future benchmarks similar to our proposed one could benefit from better controllability and removed confounders to acchieve "pure" shifts thanks to the continued progress in generative models, we argue that our approach still achieves valid and confined distribution shifts that capture the real world biases.
>
> > The approach is based on generative models, thereby introducing a real vs. synthetic distribution shift that may further influence results. A discussion - better yet: an analysis - of this likely confound is recommended.
>
> We agree that this needs to be considered carefully and fundamentally challenges any generative benchmark. To reduce the bias of the accuracy estimate between the distribution of Stable Diffusion and ImageNet, we used the ImageNet-trained text embeddings to better replicate the ImageNet distribution, which reduces the bias of the accuracy estimate by around 7%. We underline that generative benchmarks do need to address the biases arising from the real vs. synthetic shift. We use a filtering mechanism that is parameterized on a human labeled-dataset to reduce the biases of the accuracy estimates, we performed a user study to check the realism of the benchmarking images, and we compared the distribution shifts of a real-world dataset (OOD-CV) to our work. We discuss our strategies to address the realism of the generated images in the common response (iii).
>
> Does this address your comment or do you propose a different way to analyze confounder?
>
> > The paper's main claim to fame remains a bit unclear to me. [...] I recommend that the authors clearly state and justify what they believe is the primary novel contribution ("claim to fame")
>
> We really appreciate this very valuable comment and for re-framing the variety of options. We present our paper's claim to fame, i.e. our main contribution, as enabling the testing of the robustness of vision models with respect to realistic continuous-scale distribution shifts. We refer to the common response (ii) for a more elaborate discussion of the claim of fame.
>
> We additionally address your other proposals:
> While we do not focus on improving the controllability of the generative model, we aim at selecting the best methods for achieving controllability in diffusion models. In general, we would like to support that generative benchmarking can be a complementary way for benchmarking models - not reducing the importance of real-world datasets. The advantage of generative benchmarks lies in the flexibility and scalability of possible shifts, classes, and samples. Additionally, sampling from a generative models better captures the statistics of the real world than a small dataset since it is trained on very lage-scale datasets. Reducing the biases of generative benchmarks through the removal of failure cases of generative models is necessary to advance in the realm of generative benchmarking. Therefore, we propose an improved filtering mechanism in our work and show that failure cases are effectively filtered out.
>
> > If it should be a benchmark, people will want to know: who won? What's the score that I need to beat in order to be SOTA?
>
> Thank you for raising this point. We added Tab. 1 in the main paper and Tab. 2 in the supplementary where we present the mean relative corruption error as introduced for ImageNet-C. This metric allows ranking various models using one single quantity. However, underlining our key finding, we motivate that benchmarking robustness should not only involve averaged quantities.

---

> ### Author Response · Authors · 2024-11-21
> **Response 2 to reviewer zdn7**
>
> > Why were those 14 "nuisances" chosen and not others, why 14 and not, say, 50?
>
> We selected those diverse shifts as an example application of our approach, mainly inspired by ImageNet-R (8 shifts) and real-world weather shifts (6 shifts). Due to the scalable nature of our generative benchmark, our framework can be used for computing the robustness with respect to other shifts that can be expressed via a text prompt or a set of images as well. We are eager to test others shifts that the reviewers might have in mind.
>
> > What's the robustness of a failure point to random variation?
>
> We are not entirely sure if we understood this question correctly and are happy to clarify further if we did not address this point accurately: Which random variation does the reviewer refer to?
>
> Different generator seeds lead to different generated images whereas for a given seed and scale the resulting image generation is deterministic. Therefore, we assume the reviewer refers to situations where a model classifies wrongly for an earlier scale but correctly for a later scale. The failure point metric is based on the first failing scale, comparing models on individual starting images to determine when the model begins to fail. If a model correctly classified a larger scale after having failed for a smaller scale, the cumulative number of failure points would be higher than the accuracy drop.
>
> > Is performance (accuracy) always a monotonous function of the LoRA slider strength? Are there instances when that's not the case? If so, what does it mean if there are images beyond the failure point that are again correctly recognized?
>
> While the effect of the LoRA adapter is a monotonous function, it can effect the evaluated classifiers differently. Therefore, we observe few cases where a model fails first for lower scales and classifies correctly again for larger scales. Here, the failure point metrics differs from the average accuracy drop in a way that it is more sensitive to slight variations: We purposefully define this in a way that the first failure case is considered. The failure point means that there was no mis-classification before that scale.
>
> > line 43: "such approaches are not scalable" - why not? If one takes a large dataset and applies cheap corruptions like the ones from ImageNet-C, should that be considered less scaleable?
>
> ImageNet-C corruptions are very simple and fundamentally different to real-world distribution shifts.
> Our approach is different to the proposed strategy since it can achieve diverse and more realistic distribution shifts without relying on manually annotated datasets, such as (Zhao et al., 2022). Therefore, scalability not only refers to the number of classes and samples but in particular to the number and complexity of distributions shifts that can be applied.
>
> > What's the computational cost of generating the dataset?
>
> Training the LoRA adapters took 2000 GPU hours and generating the images took around 350 GPU hours. We also refer to Sec. A.5.6.
>
> > Figure 7: [...] I'd recommend using different colors here to avoid confusion.
>
> Thanks for that note. We updated the figure accordingly.

---

> > ### Comment · Reviewer_zdn7 · 2024-11-25
> > **Thanks for response**
> >
> > I'd like to thank the authors for taking the time to respond. I'm glad to see that the description of the contributions has been sharpened.
> >
> > I'd be willing to increase my score from 5 -> 6 if the authors would be open to adding the following aspects as limitations in the main paper:
> >
> > 1. Some nuisance shifts introduce confounds, e.g. "in dust" not only adds dust but also removes half of the people in the image and changes a person's shirt color from red to black. As a consequence, failures cannot always be attributed to the nuisance concept itself. I understand the author's point that this may also not be the case for other datasets but I believe this is an important limitation nonetheless, in particular when it comes to attributing failures to changes in the data.
> > 2. Acknowledging that the use of a generative model can lead to a real vs. synthetic distribution shift.
> >
> > To be clear, I don't think those limitations invalidate the approach - but I think given that they influence the interpretation of results it would be best to acknowledge them in the paper. Of course, whether to incorporate this suggestion is entirely up to the authors.

---

> ### Author Response · Authors · 2024-11-26
> **Limitations added**
>
> Thank you very much for your response.
> We agree that this point is, indeed, important to consider for people using our benchmark or researchers working on improving generative benchmarks.
>
> We added the following at the end of the method sliders section (l.232 in the updated manuscript):
> "Since our sliders do not explicitly exclude confounding variables, the applied shifts may also affect confounders inherently present due to biases in the training data. For example, as shown in \cref{fig:shift_dust}, using the \textit{in dust} slider unintentionally removes half of the people, and in \cref{fig:shift_video_game}, the background no longer represents a forest. Consequently, failures in our subsequent analysis cannot always be solely attributed to the nuisance concept itself."
>
> We also added a section about the limitations in the conclusions (l.531):
> "While our approach allows for diverse continuous nuisance shifts, it does not eliminate all confounders, meaning failures cannot always be solely attributed to the targeted nuisance concept. This highlights an inherent challenge for generative benchmarking approaches, and future advances in generative models could help mitigate these confounding factors. Additionally, while we have carefully addressed this issue in our work, we acknowledge that using generated images can lead to biases arising from the real vs. synthetic distribution shift.
>
> We hope this benchmark can encourage the community to continue working on more high-quality generative benchmarks and to adopt generated images as an additional source for systematically evaluating the robustness of vision models."
>
> Please let us know if you have other concerns.
>
> Thank you very much again for your valuable review.

---

> > ### Comment · Reviewer_zdn7 · 2024-11-26
> >
> > Thanks for your response and adding the limitations to the manuscript. I have updated my score and I'm leaning towards acceptance.

---

> ### Author Response · Authors · 2024-12-04
> **Thank you for your review**
>
> Dear reviewer zdn7,
>
> we appreciate your positive rating of our work. Thank you again for your very constructive feedback, which helped us sharpening our listed contributions.
>
> The Authors

---

### Official Review · Reviewer_LxGf · 2024-10-31

**Soundness:** 3
**Presentation:** 2
**Contribution:** 3
**Rating:** 6
**Confidence:** 5

**Summary:**

This paper introduces a benchmark, CNS-Bench, composed of synthetic images with gradual and continuous nuisance, to evaluate the robustness of classifiers in detail. The images are generated using Stable Diffusion, incorporating a wide range of individual nuisance shifts with continuous severities through LoRA adapters to diffusion models. This paper provides a detailed evaluation and analysis of various classifiers' behavior on CNS-Bench, emphasizing the advantage of utilizing generative models for benchmarking robustness across diverse continuous nuisance shifts.

**Strengths:**

1. The paper is well-motivated. Understanding the robustness of models to nuisances of varying degrees is crucial.
2. It is reasonable to generate images with gradual and continuous nuisance using Stable Diffusion and LoRA adapters.
3. The experimental section evaluates various classifiers, providing a better understanding of the robustness capabilities of these classifiers.

**Weaknesses:**

See questions.

**Questions:**

1.  I don’t understand the failure point concept in Section 3.2. This section may contain many symbols that are confusing, such as: $X_n(S), X(s_n),X_n(S_n)$ , and the subscripts  in $s_n, c_n$.
2. In Section 4, the paper mentions "activate the LoRA adapters with the selected scale for the last 75% of the noise steps". Could you provide some theoretical or empirical evidence to justify the rationale for adjusting LoRA for the last 75% of the noise steps?
3. In Section 4.2, the paper mentions "fine-tune ResNet-50 with our data and show more than 10% gains on ImageNet-R". Was the data used for fine-tuning the entire CNS-Bench or a specific style within it (such as a style closely resembling ImageNet-R distribution)? In Table 3, I noticed that after fine-tuning, the model accuracy on IN/val decreased by 2.04%. I believe the results in Table 3 do not fully support the claim regarding "the realism of generated images.”
4. For experiment about the relation between ID and OOD accuracy in section 4.3，please further elaborate on the rationale for using the slope of the linear fit between ID and OOD accuracies and the significance represented by this slope. Why not use the linear correlation coefficient？Furthermore, please provide a more detailed analysis of the results in Figure 7, particularly elucidating the impact of the strength of nuisance on the relation between ID and OOD accuracy.
5. Figures 6a and 6b evaluate the accuracy drop. I do not think this metric rational because the model size and performance on the ImageNet validation set may not necessarily align. This mismatch could result in accuracy drops of different models that are not directly comparable. Please provide the model's parameter count and the model accuracy on IN/val for reference or other evidence to claim rationality the accuracy drop.
6. Figures 4 and 5 assess using accuracy, while Figure 6 employs accuracy drop. Could you standardize to a single metric for consistency throughout the text?
7. ImageNet-C also contains images with nuisances of different strengths. What are the distinctions between CNS-Bench and ImageNet-C?
8. Could you give some experiment details of the claim “the alignment for one given seed increases in 73% for scales s > 0 for all shifts in our benchmark” in Section 3.2?

---

> ### Author Response · Authors · 2024-11-21
> **Response to reviewer LxGf**
>
> > I don’t understand the failure point concept in Section 3.2.
>
> We revised that section and we removed the indices for a more simplistic notation. We are happy to get your feedback on the updated version. In essence, the failure point distribution captures at what shift scale a classifiers fails how often. Mathematically, we define the failure point distribution through a histogram that counts the number of failure cases that occur at one of the shift scales that our benchmark considers.
>
> > Could you provide some theoretical or empirical evidence to justify the rationale for adjusting LoRA for the last 75% of the noise steps?
>
> Our goal was to maintain the coarse semantic structure of the image. Therefore, we do not activate at earlier timesteps. We mainly guided our parameter choice following Gandikota et al. (2023). Consider Fig. 13 in their supplementary for this, where they show that 75% results in a significant application of the slider concept with spatial structure preservation. Our initial experiments confirmed this. Varying this value to analyze different nuisance shifts can be an interesting step in the future.
>
> > Was the data used for fine-tuning the entire CNS-Bench? [...] after fine-tuning, the model accuracy on IN/val decreased by 2.04%.
>
> We used the entire CNS-Bench for fine-tuning. The reduced performance could be attributed to the fact that the model has to stop considering details it has learned for differentiating ImageNet classes, to improve the ImageNet-R performance. However, stylization requires less focus on the texture, which can eventually deteriorate the ImageNet performance. We refer to the common response (iii) for a discussion of the realism of the generated images.
>
> > please further elaborate on the rationale for using the slope of the linear fit between ID and OOD accuracies and the significance represented by this slope. Why not use the linear correlation coefficient?
>
> Our analysis of the relation between ID and OOD accuracy was mainly guided by the accuracy-on-the-line discussions (Miller, 2021), who find that a linear fit often captures the dependency between a ID and OOD accuracy surprisingly well. The linear slope is motivated since it quantifies which improvement of the OOD accuracy can be achieved on average when improving the ID accuracy. It helps better understanding whether an increase in OOD accuracy can be explained by a larger ID accuracy. Fig. 17 shows that most linear fits are statistically significant. To address your comment, we additionally computed the linear correlation coefficient and plot it in Fig. 18.
>
>
> > Furthermore, please provide a more detailed analysis of the results in Figure 7.
>
> The results in Fig. 7 show that the slope varies for different (1) shifts and (2) scales.
> (1) This is in line with the discussions by Miller et al. (2021): The slope varies for different datasets, i.e. the effect of a better OOD accuracy with an improved ID accuracy is not consistent across shifts.
> (2) The smaller slope for lower scales can be attributed to the subset of classes that we consider in our benchmark: A delta accuracy increase on all ImageNet classes leads to a smaller delta accuracy increase for our selected generated subset. We illustrate two examples of such a linear fit in Fig. 19. However, when the distribution shift is more prominent, an improved ID accuracy leads to a more prominent improved performance of the OOD accuracy. Our strategy allows systematically studying the effect of different distributions shifts for the relation between ID and OOD accuracy.

---

> ### Author Response · Authors · 2024-11-21
> **Response 2 to reviewer LxGf**
>
> > Figures 6a and 6b evaluate the accuracy drop. I do not think this metric rational because the model size and performance on the ImageNet validation set may not necessarily align. This mismatch could result in accuracy drops of different models that are not directly comparable. Please provide the model's parameter count and the model accuracy on IN/val for reference or other evidence to claim rationality the accuracy drop.
>
> Our benchmark mainly focuses on the accuracy drop to measure the relative OOD robustness of a model or the performance degradation for the case of nuisance shifts, following Hendrycks et al. (2018,2021). But we agree that there exists a relation between ID and OOD accuracy. We considered and reported the relation between ID and OOD accuracy in Fig. 7. To additionally address your comment, we report the ImageNet validation accuracies and the model parameter counts in Tab. 5.
> While the model parameter count purposefully varies for the *model size* axis, we use the same ViT-B backbone for the *pre-training* axis, and a comparable number of parameters for the *architecture* axis. To further address your question, we additionally compute the accuracy gain after subtracting the effect of an improved OOD accuracy taking into account the linear fit that we discussed previously. We plot the results along the model architecture axis in Fig. 19. We find that ConvNext achieves the best accuracy gain after removing the linear dependency of an improved ID accuracy. However, we underline that this measure needs to be considered cautiously since the (1) linear fit depends on the selected models for computing the statistics and (2) the linear fit might not always describe the relationship sufficiently well. Consider, e.g., the discussions in '_Accuracy on the Curve: On the Nonlinear Correlation of ML Performance Between Data Subpopulations_' by W. Liang (2023) or in 'Accuracy on the wrong line: On the pitfalls of noisy data for out-of-distribution generalisation' by A. Sanyal (2024).
> Therefore, we would rather argue for consistently using the accuracy drop as a measure of robustness. We are happy to get your feedback on this point.
>
> > Figures 4 and 5 assess using accuracy, while Figure 6 employs accuracy drop. Could you standardize to a single metric for consistency throughout the text?
>
> Thanks for pointing out that inconsistency. We updated the figures accordingly.
>
> > What are the distinctions between CNS-Bench and ImageNet-C?
>
> Imagenet-C does not contain real-world shifts but only simple corruptions, which is a fundamentally different nuisance shift, not capturing all realistic distributions shifts.
> We also refer to the discussion in the general comment (i).
>
> > Could you give some experiment details of the claim “the alignment for one given seed increases in 73% for scales s > 0 for all shifts in our benchmark” in Section 3.2?
>
> To evaluate whether the application of our sliders yields an increase of the desired shift, we compute the CLIP alignment of the generated image to the text prompt describing the shift. Increasing the scale $s$ by 0.5 increases the CLIP alignment in 73% of the cases. This shows that the shift is increased in the majority of the cases when relying on the CLIP alignment score. An explanation for cases where the CLIP score does not increase can be that the applied shift by the LoRA slider does not exhibit the characteristics that CLIP measures for that shift although the change can be visible for a human. We illustrate such a case in Fig. 16, where the painting shift can be observed but the CLIP alignment to the shift drops.
> We moved this part to Sec. A.5.4 in the supplementary and added explanations there.

---

> > ### Comment · Reviewer_LxGf · 2024-11-25
> >
> > Thanks for your detail rebuttal, I think most of my concerns are solved, I will change the score.

---

> ### Author Response · Authors · 2024-12-04
> **Thank you for your review**
>
> Dear reviewer LxGf,
>
> thank you for recognizing the significance of our contributions and updating your score. Thank you again for your constructive feedback, which helped improve the clarity of our paper.
>
> The Authors

---

### Official Review · Reviewer_xh13 · 2024-11-03

**Soundness:** 2
**Presentation:** 2
**Contribution:** 3
**Rating:** 6
**Confidence:** 3

**Summary:**

This paper introduces a novel benchmark dataset for evaluating model robustness by systematically controlling individual nuisance factors. The dataset allows for a precise assessment of the failure points of vision models, based on the severity of these controlled nuisance factors. The authors find that model rankings vary with changes in shift severity, and model architecture is a key factor in robustness.

**Strengths:**

Instead of measuring average accuracy drop across all nuisance shifts, the authors consider evaluating model performance at specific levels of nuisance shifts, enabling a detailed analysis of failure points in vision models.

**Weaknesses:**

1. Unclear contributions: The contributions listed in the paper seem overlapping. The distinctions among them are insufficiently clear. Notably, the third contribution is not visible in the main text. While the paper claims 14 distinct nuisance shifts as a key contribution, it lacks an explanation or rationale for selecting these specific shifts. Since this is a foundational aspect of the contribution, detailed descriptions should be provided in the main text, not relegated to the appendix.

2. Ambiguity in benchmark superiority: The authors assert that their benchmark outperforms existing benchmarks for evaluating model robustness by incorporating nuisance shifts across multiple severity levels. However, earlier works by Hendrycks & Dietterich (2018) and Kar et al. (2022) already support multi-severity analysis for vision model failure points. Thus, the authors should clarify how their benchmark framework distinctly advances beyond these existing approaches.

3. Inconsistent statements on model robustness: In line 451, the authors claim that transformers are more robust than CNNs, yet this statement seems contradicted by Fig. 6a, where ConvNext outperforms ViT and DeiT but performs slightly worse than DeiT3. This inconsistency suggests that CNNs may not always be less robust than transformers, and the statement should be re-evaluated or clarified.

4. Validation of realistic nuisance shifts: While the authors argue that the benchmark includes realistic nuisance shifts, the realism of these diffusion-generated images is not substantiated. Proper validation, such as human assessment, would enhance the credibility of this claim.

5. Readability of figures: The font size in several figures is too small, which detracts from readability. Increasing the font size would improve clarity for readers.

**Questions:**

1. Self-supervised pre-training: Why is DINOv1 using linear probing compared with other models? This seems to create an unfair comparison, as linear probing may not fully reflect the robustness of self-supervised models relative to other models in the evaluation. Could you clarify the rationale behind this comparison approach?

---

> ### Author Response · Authors · 2024-11-21
> **Response to reviewer xh13**
>
> > Unclear contributions: The contributions listed in the paper seem overlapping.
>
> Thank you for pointing out this. We re-structured our contributions differently to reduce the overlap and we refer to the common response (ii). The new structure is that we separate (1) the framework for continuous shifts, (2) the OOC filtering strategy, and (3) the evaluation and analysis of various models.
>
> > While the paper claims 14 distinct nuisance shifts as a key contribution, it lacks an explanation or rationale for selecting these specific shifts.
>
> We added the motivation for the 14 shifts in the introduction and in the experimental details. In short, we selected those diverse shifts as an example application of our approach, mainly inspired by ImageNet-R (8 shifts) and real-world weather shifts (6 shifts). Due to the scalable nature of our generative benchmark, our framework can be used for computing the robustness with respect to other shifts that can be expressed via a text prompt or a set of images as well. We are eager to test others shifts that the reviewers might have in mind.
>
> > Ambiguity in benchmark superiority : The authors assert that their benchmark outperforms existing benchmarks for evaluating model robustness by incorporating nuisance shifts across multiple severity levels. However, earlier works by Hendrycks & Dietterich (2018) and Kar et al. (2022) already support multi-severity analysis for vision model failure points. Thus, the authors should clarify how their benchmark framework distinctly advances beyond these existing approaches.
>
> While the mentioned works allow multiple severity levels, they are restricted to synthetic or simple semantic corruptions. Our approach allows real-world distribution shifts and can be easily scaled to other shifts. Applying diverse and realistic natural real-world shifts is fundamentally more challenging and motivates the application of generative models. We do not consider our work as superior but complementary to the benchmarks that support simple multi-scale corruptions.
>
> We also refer to our common response (i).
>
> > Inconsistent statements on model robustness: In line 451, the authors claim that transformers are more robust than CNNs, yet this statement seems contradicted by Fig. 6a, where ConvNext outperforms ViT and DeiT but performs slightly worse than DeiT3. This inconsistency suggests that CNNs may not always be less robust than transformers, and the statement should be re-evaluated or clarified.
>
> We believe there may be a misunderstanding. Our findings indicate that CNNs generally perform worse than transformers when using modern training recipes like DeiT3, which we also tried to communicate. We noted in the paper: "A modern CNN (ConvNext) outperforms transformers but is less robust than when using modern training recipes (DeiT3)." (l.451 in the original version). We apologize for the confusion and we reformulated that paragraph for clarity.
>
> > Validation of realistic nuisance shifts: While the authors argue that the benchmark includes realistic nuisance shifts, the realism of these diffusion-generated images is not substantiated. Proper validation, such as human assessment, would enhance the credibility of this claim.
>
> Thank you for raising that important point. We refer to the common response (iii) for a discussion of our strategies to account for the realism. Specifically, following your advice, we also validated the realism of our images through human assessment, which we document in Sec. A.7. One percent of our benchmarking samples were detected to be not a sample of the desired class according to our user study where each image was checked by two different individuals.
>
> > Readability of figures: The font size in several figures is too small, which detracts from readability. Increasing the font size would improve clarity for readers.
>
> Thank you for that comment. We updated the figures accordingly.
>
> > Self-supervised pre-training: Why is DINOv1 using linear probing compared with other models? This seems to create an unfair comparison, as linear probing may not fully reflect the robustness of self-supervised models relative to other models in the evaluation. Could you clarify the rationale behind this comparison approach?
>
> Originally, we only compared using the publicly available linear-probed DINOv1 model since no fine-tuned model was available. To address your comment, we fine-tuned DINOv1 and updated the results and discussions accordingly (consider, e.g., Tab. 1,2,3 and Fig. 6). The fine-tuned DINOv1 model achieves a comparable performance to MoCov3 on CNS-Bench, with a better performance on small scales but achieving slightly worse results on larger shift scales.

---

> > ### Comment · Reviewer_xh13 · 2024-11-26
> >
> > Thank you for the detailed responses. I think my main concerns regarding the contributions and technical details of this work were addressed. I will update the score.

---

> ### Author Response · Authors · 2024-12-04
> **Thank you for your review**
>
> Dear reviewer xh13,
>
> thank you for recognizing the significance of our contributions and updating your score. Thank you again for your critical and constructive feedback, which helped us more clearly state the relevance of our benchmark.
>
> The Authors

---

### Official Review · Reviewer_8WvD · 2024-11-03

**Soundness:** 3
**Presentation:** 3
**Contribution:** 3
**Rating:** 6
**Confidence:** 3

**Summary:**

The paper introduces CNS-Bench, which uses generative models for benchmarking robustness across diverse continuous nuisance shifts by applying LoRA adapters to diffusion models.

**Strengths:**

- The paper is well-structured, and the proposed CNS-Bench benchmark is simple yet effective for evaluating model robustness. The authors provide comprehensive discussions along three key dimensions—architecture, number of parameters, and pre-training paradigm—giving clear insights into the paper's findings.
- In addition to the proposed dataset for benchmarking model robustness, the authors present an annotated dataset to benchmark OOC) filtering strategies. They introduce a novel filtering mechanism that significantly improves filter accuracy, which is a notable contribution.
- The application of LoRA sliders to compute shift levels continuously is a particularly innovative and inspiring approach. This adds an interesting methodological contribution to the paper.

**Weaknesses:**

- The novelty of the insights presented in this paper could be more compelling. For example, in Figure 6, are there any underlying reasons or mechanisms that could provide a deeper understanding of the results? It would be beneficial to explore these further to add depth to the conclusions.

**Questions:**

There is a lack of clarity on how the dataset handles potentially unrealistic or counter-intuitive scenarios, such as cars driving on water. How are these cases addressed? A discussion on the handling of such edge cases would improve the comprehensiveness of the dataset.

---

> ### Author Response · Authors · 2024-11-21
> **Response to reviewer 8WvD**
>
> > The novelty of the insights presented in this paper could be more compelling. For example, in Figure 6, are there any underlying reasons or mechanisms that could provide a deeper understanding of the results?
>
> Thank your for pointing out that our discussions were not clearly stating the novelty of our observations. We try to formulate the key insight of our benchmark in the following more clearly.
>
> First of all, we find that there is not a single model that rules all realistic shifts and scales equally well.
> When averaging the performance over all shifts, model rankings do not heavily change for different scales (Fig. 6a). This states that a robust model that is robust on a weakly-shifted OOD dataset A tends to be robust as well on heavily-shifted dataset B. However, considering this average metric is not sufficient to evaluate the robustness of a model on a specific OOD scenario: When comparing the performances for individual nuisance shifts, the model rankings can significantly change for different scales. This effect depends on the considered shift and models. E.g., we observe that the effect of weather variations, such as rain or fog, results in varying performance for different scales and shifts but increasing the scale does not significantly change the model rankings (Fig. 9). This is similar to ImageNet-C, where different corruptions lead to different performance drops over varying scales (Fig. 22). However, some style changes impact the models clearly differently for various scales (Fig. 6b and 9). We believe that this might be attributed to the effect that different models focus on different characteristics of the classes, which are modulated differently at different scales of the shift, which, we think, is a note-worthy novel finding.
>
> We hope, this addressed your comment appropriately. Depending on your feedback, we will update the discussions in the main paper accordingly.
>
> > There is a lack of clarity on how the dataset handles potentially unrealistic or counter-intuitive scenarios, such as cars driving on water.
>
> We believe there are two points to address this question. Our benchmark follows the statistics of the training data of the generative model, which captures the distribution of available images. Unrealistic cases can however still happen but it is typically hard to automatically differentiate between edge cases and physically implausible or unrealistic cases. Consider, e.g., the presented examples in Fig. 30 that relate to the example of cars driving on water.
>
> Nevertheless, it could be argued whether such unrealistic or counter-intuitive scenarios are problematic as long as the class is still recognizable since humans also generalize to edge cases. It might be even a motivation to use generative models for benchmarking to generate edge cases that only rarely occur in real world scenarios. Thank you for raising this point.
>
> We hope our answer addressed your question appropriately. Please feel free to follow up if you have any remaining questions or concerns.

---

> ### Author Response · Authors · 2024-12-04
> **Thank you for your review**
>
> Dear reviewer 8WvD,
>
> we hope, our rebuttal addressed your concerns. Thank you again for your positive review and for recognizing the significance of our work.
>
> The Authors

---

### Author Response · Authors · 2024-11-21
**Common response**

We thank all reviewers for their questions and constructive feedback. We are pleased that the reviewers recognize the "particularly innovative and inspiring approach" (8WvD), that "understanding the robustness of models to nuisances of varying degrees is crucial" (LxGf), that our "technical approach is a nice, simple and creative application of generative diffusion models" (zdn7), that our strategy is "technically sounds" (1ik7), and that the reviewers "appreciate the thorough analysis and sanity-checks" (1ik7).

In the following, we discuss common questions among the reviewers.

> **(i)** Why is a benchmark supporting multiple scales relevant (1ik7) and why is our benchmark superior to ImageNet-C? (xh13,LxGf)

ImageNet-C, the pioneering work by Hendrycks & Dietterich (2018) has shown that it is important to consider multiple corruption scales. In particular, they showed that it is possible that a classifier A has a lower performance drop than classifier B, even though classifier A degrades more gracefully in the presence of corruptions, and hence might be preferable over classifiers that degrade suddenly. However, ImageNet-C and, similarly, 3D-CC by Kar et al. (2022) are restricted to synthetic or simple semantic corruptions and do not consider a variety of real-world distribution shifts. The motivation for considering multiple scales also holds for real-world nuisance shifts. Enabling a systematic of real-world nuisance shifts at multiple scales is the focus of our work.


> **(ii)** Claim to fame (zdn7) and unclear contributions (xh13)

We present our paper's claim to fame, i.e. our main contribution, as follows:
CNS-Bench allows testing the robustness of vision models with respect to fine-granular and continuous-scale real-world distribution shifts for the first time. This significantly extends the variety and diversity of robustness evaluations at multiple scales compared to ImageNet-C.

Our main new finding is: There is not one single model that performs best on all distribution shifts and the model rankings can vary for different shifts and scales. Therefore, the selection of the considered shifts and their severities clearly influence the final model rankings when comparing averaged performances. Consequently, we underline the importance of applying nuisance shifts that are more specific to an OOD scenario of interest.

Further, we address an urging challenge for generative benchmarking by proposing an improved filtering mechanism for removing failed generated images.

Since CNS-Bench allows applying continuous shifts, it also enables the computation of failure points for diverse distribution shifts that go beyond the analysis of simple corruptions.

Our approach advances the field since we allow a more nuanced study of model robustness by applying controlled multiple-scale real-world distribution shifts.

We followed the advice by reviewer xh1303 and restructured the list of contributions and updated the manuscript accordingly: 1) Benchmark for continuous shifts, 2) filtering strategy, 3) systematic evaluation.


> **(iii)** Realism of the generated images (xh13,LxGf)

In our work, we address the realism of the generated images in various ways:

- First, we proposed an improved filtering mechanism to remove OOC samples. To parameterize the filtering, we collected a large manually labeled dataset. We illustrated that our automatically filtered dataset results in comparable accuracy estimate as the manually labeled dataset.
- Second, we purposefully performed the comparison with the OOD-CV dataset to compare the effect of real-world distribution shifts and our generative approach.
- Third, we fine-tuned a classifer on our data and achieved improved performance on the real-world IN-R dataset.
- Fourth, following the advice by reviewer xh13, we also conducted a user study to evaluate whether our filtered dataset contains images that do not represent the class and we discuss the results in Sec. A.7: The estimated ratio of out-of-class samples equals 1% with a margin of error of 0.5% for a one-sigma interval.


> Updating the manuscript

We started updating the manuscript according to your feedback, added new results, supporting figures and tables, and we will further improve it taking into account the reviewer's reactions on our rebuttal.
In our answers, we provide references to the updated manuscript if not stated differently.

---

### Meta-Review · Area_Chair_26Tg · 2024-12-21

**Metareview:**

The manuscript introduces CNS-Bench, a benchmarking framework utilizing LoRA adapters and diffusion models to evaluate vision model robustness under continuous nuisance shifts. While the paper is technically sound and offers a creative application of generative models, it does not provide compelling evidence of substantial novelty or practical utility. Below are key points summarized from reviewer feedback:

Strengths:
* Technical Soundness: The application of LoRA adapters and generative models is well-executed, with thorough evaluations of generated image quality and classifier robustness.
* Innovative Approach: The paper proposes a unique method for realizing continuous nuisance shifts using generative models, which could inspire further research.
* Detailed Analysis: The authors conduct robust experiments, including failure point analyses, and address feedback with significant manuscript improvements.

Weaknesses:
* Limited Novelty: The methodology primarily combines existing techniques without substantial innovation. The benchmark does not surpass existing alternatives like ImageNet-C in terms of practical utility or realism. Continuous nuisance shifts, while novel in implementation, do not demonstrate significant new insights or impact.
* Utility and Scope: The practical relevance of the benchmark is questionable, given modest performance drops even for challenging nuisance shifts. The generated images often fail to emulate realistic OOD scenarios, particularly for weather-based shifts like fog and rain.
* Lack of Clear Impact: The paper fails to articulate a strong "claim to fame" or compelling use case. Model rankings remain largely consistent across scales, reducing the value of continuous shift evaluation.
* Presentation Issues: The manuscript lacks clarity in critical sections (e.g., methodology). Figures and metrics (e.g., accuracy drop) were initially unclear, though these were partially addressed during the rebuttal.

While the manuscript demonstrates solid technical work and engages effectively with reviewer feedback, it lacks the novelty, clarity, and demonstrated utility required for publication at ICLR. The benchmark’s incremental contribution and limited practical relevance suggest that it is not ready for acceptance in its current form.

**Additional Comments On Reviewer Discussion:**

Key Points Raised by Reviewers and Authors’ Responses:

Novelty and Utility:
* Point: Limited innovation; unclear practical utility of continuous shifts.
* Response: Authors argued that the benchmark offers new ways to analyze robustness with realistic nuisance shifts. Highlighted insights on model ranking changes across scales.
Outcome: Reviewers remained unconvinced about practical impact, citing marginal utility.

Realism and Quality of Generated Images:
* Point: Images (e.g., rain, fog) do not represent challenging OOD scenarios; confounding effects noted in generated images.
* Response: Authors acknowledged limitations and added discussions about biases and confounders in generative models.
* Outcome: Reviewers appreciated acknowledgment but emphasized limited relevance of such shifts.

Clarity and Presentation:
* Point: Poor clarity in methodology (e.g., failure points, calibration of shifts).
* Response: Authors revised key sections, improved figure captions, and added confidence intervals to results.
* Outcome: Reviewers noted improvements but retained concerns about the benchmark’s overall coherence.

Benchmark Evaluation Metrics:
* Point: Questions on relevance of metrics like accuracy drop and failure points.
* Response: Clarified metrics and removed averaged failure points across shifts.
* Outcome: Clarifications addressed confusion but did not substantially change reviewers’ impressions.

Scope of Contribution:
* Point: Benchmark’s contributions overlap with existing methods (e.g., ImageNet-C).
* Response: Authors emphasized scalability and unique application of LoRA adapters.
* Outcome: Reviewers accepted this as a methodological contribution but viewed it as incremental.

---

### Decision · Program_Chairs · 2025-01-22

Reject